# Happy: A Debiased Learning Framework for Continual Generalized Category Discovery

**Shijie Ma**[1,2]**, Fei Zhu**[3]**, Zhun Zhong**[4,5]**, Wenzhuo Liu**[1,2]**, Xu-Yao Zhang**[1,2]***, Cheng-Lin Liu**[1,2]

[1]MAIS, Institute of Automation, Chinese Academy of Sciences, China
[2]School of Artificial Intelligence, University of Chinese Academy of Sciences, China
[3]Centre for Artificial Intelligence and Robotics, HKISI-CAS, China
[4]School of Computer Science and Information Engineering, Hefei University of Technology, China
[5]School of Computer Science, University of Nottingham, NG8 1BB Nottingham, UK
`mashijie2021@ia.ac.cn`    `xyz@nlpr.ia.ac.cn`

## Abstract

Constantly discovering novel concepts is crucial in evolving environments. This paper explores the underexplored task of Continual Generalized Category Discovery (C-GCD), which aims to incrementally discover new classes from *unlabeled* data while maintaining the ability to recognize previously learned classes. Although several settings are proposed to study the C-GCD task, they have limitations that do not reflect real-world scenarios. We thus study a more practical C-GCD setting, which includes more new classes to be discovered over a longer period, without storing samples of past classes. In C-GCD, the model is initially trained on labeled data of known classes, followed by multiple incremental stages where the model is fed with unlabeled data containing both old and new classes. The core challenge involves two conflicting objectives: discover new classes and prevent forgetting old ones. We delve into the conflicts and identify that models are susceptible to *prediction bias* and *hardness bias*. To address these issues, we introduce a debiased learning framework, namely Happy, characterized by **H**ardness-**a**ware **p**rototype sampling and soft entro**py** regularization. For the *prediction bias*, we first introduce clustering-guided initialization to provide robust features. In addition, we propose soft entropy regularization to assign appropriate probabilities to new classes, which can significantly enhance the clustering performance of new classes. For the *harness bias*, we present the hardness-aware prototype sampling, which can effectively reduce the forgetting issue for previously seen classes, especially for difficult classes. Experimental results demonstrate our method proficiently manages the conflicts of C-GCD and achieves remarkable performance across various datasets, *e.g.*, 7.5% overall gains on ImageNet-100. Our code is publicly available at https://github.com/mashijie1028/Happy-CGCD.

## 1  Introduction

In the open world [1, 2, 3], visual concepts are infinite and evolving and humans can cluster them with previous knowledge. It is also important to endow AI with such abilities. In this regard, Novel Category Discovery (NCD) [4, 5, 6] and Generalized Category Discovery (GCD) [7, 8, 1, 9, 10] endeavor to transfer [4, 11] the knowledge from labeled classes to facilitate clustering new classes. However, they are constrained to *static* settings where models only learn *once*, which contradicts the ever-changing world. Thus, extending them to the temporal dimension is important. In the literature, Continual Novel Category Discovery (C-NCD) [12, 13, 14] and Continual Generalized Category

---

*Corresponding author.

38th Conference on Neural Information Processing Systems (NeurIPS 2024).

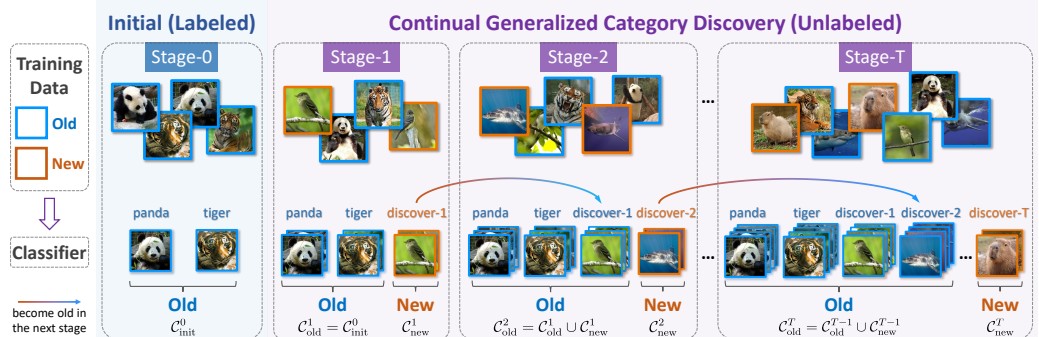

Figure 1: The diagram of Continual Generalized Category Discovery (C-GCD). In this paper, we focus on a more pragmatic setting with (1) more continual stages and more novel categories, (2) rehearsal-free learning, and (3) no prior knowledge of the ratio of new class samples.

Discovery (C-GCD) [15, 16, 17, 18] aim to discover novel classes continually. C-NCD assumes all data come from new classes, while C-GCD further considers the coexistence of old and new ones. However, C-GCD still has some limitations, *e.g.*, some works [15, 17] store labeled data of old classes, causing storage and privacy [19] issues. Others [16, 18] consider limited incremental stages and novel categories or assume a prior ratio of known samples [15], failing to reflect practical cases.

In this paper, we tackle the task of C-GCD, but with more realistic considerations: (1) More learning stages with more new classes. (2) At each stage, data from previous stages are inaccessible [20] for storage and privacy concerns. (3) Unlabeled data contain samples from old classes but are fewer than new ones in each class, and the ratio of them is unknown. The C-GCD setting is illustrated in Figure 1 with two phases: (1) Initial supervised learning (Stage-0). The model is trained on labeled classes to acquire general knowledge. (2) Continual unsupervised discovery (Stage-1 $\sim T$). At each stage, the model learns from unlabeled data containing both new and old classes. Note that, old classes include initially labeled classes as well as those discovered in previous stages. The core challenge is managing the conflicts between discovering new classes and preventing forgetting old ones.

To explore the nature of the conflicts, we conducted preliminary experiments (Section 3.2) which reveal two issues: Models (1) tend to misclassify new classes as old, leading to collapsed accuracy of new classes, and (2) exhibit catastrophic forgetting of old classes. We summarize them as underlying issues to be addressed: (1) Models display overconfidence in old classes and severe *prediction bias*. (2) The features of old classes are disrupted when learning novel classes. Meanwhile, the similarity between clusters varies, leading to biased hardness across classes for classification.

To address these issues, we propose a debiased framework, namely Happy, which is characterized by **H**ardness-**a**ware **p**rototype sampling and soft entro**py** regularization. Specifically, on the one hand, to better discover new classes during incremental stages, we utilize clustering-guided initialization for new classes, ensuring a reliable feature distribution. More importantly, to mitigate the *prediction bias* between new and old classes, we introduce soft entropy regularization to allocate necessary probabilities to the classification head of new classes, which is essential for new class discovery. On the other hand, to prevent catastrophic forgetting in rehearsal-free C-GCD, we model the class-wise distribution in the feature space for old classes, and sample them when learning novel classes, which significantly mitigates catastrophic forgetting. Furthermore, we devise a metric to quantify the hardness of each learned class, and prioritize sampling features from categories with greater difficulty. This helps the model to consolidate difficult knowledge accordingly and thus improves the overall performance. Consequently, these designs enable our model to specifically address the challenges in C-GCD, *i.e.*, effectively discover new classes while preventing catastrophic forgetting of old classes.

In summary, our contributions are: (1) We extend Continual Generalized Category Discovery (C-GCD) to realistic scenarios. In addition, we propose a debiased learning framework called Happy, which excels in effectively discovering new classes while preventing catastrophic forgetting with reduced bias in the introduced C-GCD settings. (2) We propose cluster-guided initialization and soft entropy regularization for collectively ensuring stable clustering of new classes. On the other hand, we present hardness-aware prototype sampling to mitigate forgetting. (3) Comprehensive experiments show that our method remarkably discovers new classes with minimal forgetting of old classes, and outperforms state-of-the-art methods by a large margin across datasets.

## 2 Related Works

**Category Discovery.** Novel Category Discovery (NCD) [4, 5, 21] is firstly formalized as deep transfer clustering [4], *i.e.*, transferring the knowledge from labeled classes to help cluster new ones. Early works employ robust rank statistics [5, 22] for knowledge transfer. UNO [6] proposes a unified objective with Shinkhorn-Knopp algorithm [23]. Later works [24, 25, 26] exploit relationships between samples and classes. NCD assumes unlabeled data only contain new classes. Instead, Generalized Class Discovery (GCD) [7, 27] further permits the existence of old classes. Thus models need to classify old classes and cluster new ones in the unlabeled data. Recent works handle GCD with non-parametric contrastive learning [8, 28, 10] or parametric classifiers with self-training [9, 29, 30]. More recent works explore GCD in other settings, *e.g.*, active learning [31] and federated learning [32]. In summary, both NCD and GCD are limited to *static* settings where models only learn *once*.

**Continual Category Discovery.** Pioneer works [12, 13, 14] study the incremental version of NCD, assuming unlabeled data only contain new classes, and we call them C-NCD. Recent works [15, 16, 17, 18] explore the incremental version of GCD, we collectively refer to them as Continual Generalized Category Discovery (C-GCD). GM [15] proposes a framework of growing and merging. In the growing phase, the model performs novelty detection and implements clustering on the novelties. Then GM integrates the newly acquired knowledge with the previous model in the merging stage. Kim *et al.* [16] utilize noisy label learning and the proxy and anchor scheme to split the data in C-GCD. Zhao *et al.* [17] propose a non-parametric soft nearest-neighbor classifier and a density-based sample selection method. Orthogonally, Wu *et al.* [18] argue that the initial labeled data are not fully exploited and present a meta-learning [33] framework to learn a better initialization for continual discovery. Despite effectiveness, C-GCD settings studied by the above methods still have some limitations, *e.g.*, the number of stages is very few with limited new classes, and the assumption of prior ratio of old classes or storing previously labeled samples is unrealistic [19, 34]. In this paper, we extend C-GCD to more pragmatic scenarios, as shown in Figure 1.

## 3 Preliminaries

We first formalize Continual Generalized Category Discovery (C-GCD) (Section 3.1). To delve deeper into the issues, we conduct preliminary experiments (Section 3.2). Results reveal that models are susceptible to two types of *bias*, which significantly degrade the performance and motivate us to propose the debiased learning framework in Section 4.

### 3.1 Problem Formulation and Notations

**Task Definition.** As shown in Figure 1, C-GCD has two phases: (1) Initial supervised learning (Stage-0). The model is trained on labeled data $\mathcal{D}_{\text{train}}^0 = \{(\boldsymbol{x}_i^l, y_i)\}_{i=1}^{N^0}$ of initially labeled classes $\mathcal{C}_{\text{old}}^0 = \mathcal{C}_{\text{init}}^0$, to learn general knowledge and representations. We denote $\mathcal{C}_{\text{new}}^0 = \text{None}$. (2) Continual unsupervised discovery (Stage-1 $\sim T$). At Stage-$t$ ($1 \leq t \leq T$), the model is fed with an unlabeled dataset $\mathcal{D}_{\text{train}}^t = \{(\boldsymbol{x}_i^u)\}_{i=1}^{N^t}$, which contains both old and new classes. We denote the categories in $\mathcal{D}_{\text{train}}^t$ as $\mathcal{C}^t = \mathcal{C}_{\text{old}}^t \cup \mathcal{C}_{\text{new}}^t$. $K_{\text{old}}^t = |\mathcal{C}_{\text{old}}^t|$, $K_{\text{new}}^t = |\mathcal{C}_{\text{new}}^t|$ and $K^t = K_{\text{old}}^t + K_{\text{new}}^t$ denote the number of "old", "new" and "all" classes respectively. Note that, after the first stage, *i.e.*, when $t \geq 2$, "old" classes include initially labeled classes $\mathcal{C}_{\text{init}}^0$ and all new classes discovered in previous stages, *i.e.*, $\mathcal{C}_{\text{old}}^t = \mathcal{C}_{\text{init}}^0 \cup \{\mathcal{C}_{\text{new}}^i\}_{i=1}^{t-1}$, and "new" classes refer to the classes unseen before. At the next stage, new classes from the current stage become the subset of old classes, *i.e.*, $\mathcal{C}_{\text{old}}^t = \mathcal{C}_{\text{old}}^{t-1} \cup \mathcal{C}_{\text{new}}^{t-1}$. The number of novel classes $K_{\text{new}}^t$ at stage $t$ is known *a-prior* or estimated using off-the-shelf methods [5, 7, 35] in advance. After training of each stage, the model will be evaluated on the disjoint test set $\mathcal{D}_{\text{test}}^t = \{(\boldsymbol{x}_i, y_i)\}_{i=1}^{N_{\text{test}}^t}$ containing all seen classes $\mathcal{C}_{\text{old}}^t \cup \mathcal{C}_{\text{new}}^t$.

**Realistic Considerations.** Our C-GCD is more realistic than prior arts [15, 16, 18] in that: (1) More stages with more new classes to be discovered. (2) Rehearsal-free. Previous samples are inaccessible for storage and privacy issues. (3) At each continual stage, old classes have fewer samples per class than new classes in the unlabeled data, and the proportion of old samples is unknown.

**Notations.** At Stage-$t$, we decompose the model into encoder $\boldsymbol{f}_\theta^t(\cdot)$ and parametric classifier $g_\phi^t = [\{\phi_i^{\text{old}}\}_{i=1}^{K_{\text{old}}^t}; \{\phi_j^{\text{new}}\}_{j=1}^{K_{\text{new}}^t}]$ with head of old and new classes. The classifier is $\ell_2$-normalized without bias term, *i.e.*, $\|\phi_i^t\| = 1$. The encoder maps the input $\boldsymbol{x}_i$ to a feature vector $\boldsymbol{z}_i = \boldsymbol{f}_\theta^t(\boldsymbol{x}_i) \in \mathbb{R}^d$. Here,

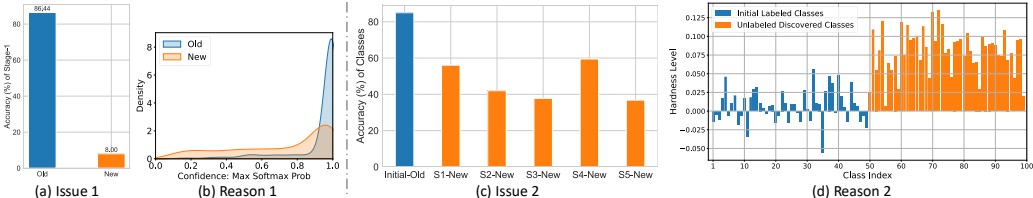

Figure 2: Preliminary results. We identify two issues and underlying causes, including **(a) Issue 1:** performance gap between old and new classes, caused by **(b) Reason 1:** model's overconfidence in old classes, *i.e.*, *prediction bias*. **(c) Issue 2:** accuracy fluctuations in new class across various stages, caused by **(d) Reason 2:** different categories have varying levels of difficulty, *i.e.*, *hardness bias*.

we use $\ell_2$-normalized hyperspectral feature space, *i.e.*, $z_i = z_i / \|z_i\|$. The classifier finally produces a probability distribution $p_i = \sigma(g_\phi^t(z_i)/\tau_p) \in \mathbb{R}^{K^t}$ using softmax function $\sigma(\cdot)$.

### 3.2 Preliminary Experiments: Two Bias Problems

We conduct preliminary experiments on CIFAR100 [36] using the model described in Section 3.1, which is initially trained on $\mathcal{D}_{\text{train}}^0$ and continually discovers new classes on $\mathcal{D}_{\text{train}}^t$ using unsupervised self-training scheme [9]. Results reveal that models are prone to the following two types of *bias*.

***Prediction bias* in probability space.** As illustrated in Figure 2 (a), the model's accuracy for new classes has collapsed. The reason is that old classes $\mathcal{C}_{\text{old}}^0$ are trained under full supervision while new classes are under unsupervised self-training [9, 37], which brings about overconfidence [38, 39, 40] in old classes, as in (b). In this case, *prediction bias* could occur, where some new classes are incorrectly predicted as old ones, which motivates us to constrain the model to give necessary attention and predictive probabilities to new classes to compensate for this intrinsic gap, as discussed in Section 4.2.

***Hardness bias* in feature space.** After adding constraints to ensure learning new classes (Section 4.2), their accuracies significantly fluctuate across incremental stages, leading to unstable performance, as shown in Figure 2 (c). The underlying cause is that some clusters are more similar to others in the feature space, resulting in lower accuracy of these difficult classes. As in (d), *hardness bias* (defined in Section 4.3) is obvious across classes. This paper focuses on the hardness of previously learned categories $\mathcal{C}_{\text{old}}^t$, and addresses how to avoid these biases in preventing forgetting in Section 4.3.

## 4 The Proposed Framework: Happy

**Overview of the Method.** As shown in Figure 1, C-GCD has two phases: (1) Initial supervised learning (Stage-0). The model is trained on labeled samples of $\mathcal{C}_{\text{init}}^0$ (Section 4.1). Our contribution mainly lies in (2) Continual unsupervised discovery (Stage-1~ T). Motivated by the conflicts between new class discovery and the forgetting of old classes, as well as the two types of *bias* discussed in Section 3.2, we propose the debiased learning framework Happy as illustrated in Figure 3. Specifically, for category discovery, we propose initialization of new heads and soft entropy regularization to resist *prediction bias* (Section 4.2). To mitigate forgetting, we consider *hardness bias* and present hardness-aware prototype sampling (Section 4.3). The overall objective is derived in Section 4.4.

### 4.1 Supervised Training at the Initial Stage

At Stage-0, the model is trained on labeled data $\mathcal{D}_{\text{train}}^0$ from a large number of classes $\mathcal{C}_{\text{init}}^0$ to learn general representations, which serves as the foundation for subsequent continual category discovery. We use standard supervised cross-entropy loss on the batch $B$: $\mathcal{L}_{\text{cls}} = \frac{1}{|B|} \sum_{i \in B} -y_i \log p_i$, where $p_i = \sigma(g_\phi^0(f_\theta^0(x_i))/\tau)$ denotes the prediction. To reduce overfitting, we further employ supervised [41] and self-supervised contrastive learning [42] in the $\ell_2$-normalized projection space:

$$\mathcal{L}_{\text{con}}^l = -\frac{1}{|B|} \sum_{i \in B} \frac{1}{|\mathcal{P}(i)|} \sum_{q \in \mathcal{P}(i)} \log \frac{\exp(h_i^\top h_q'/\tau_c)}{\sum_{n \neq i} \exp(h_i^\top h_n'/\tau_c)}, \ \mathcal{L}_{\text{con}}^u = -\frac{1}{|B|} \sum_{i \in B} \log \frac{\exp(h_i^\top h_i'/\tau_c)}{\sum_{n \neq i} \exp(h_i^\top h_n'/\tau_c)},$$
$$(1)$$

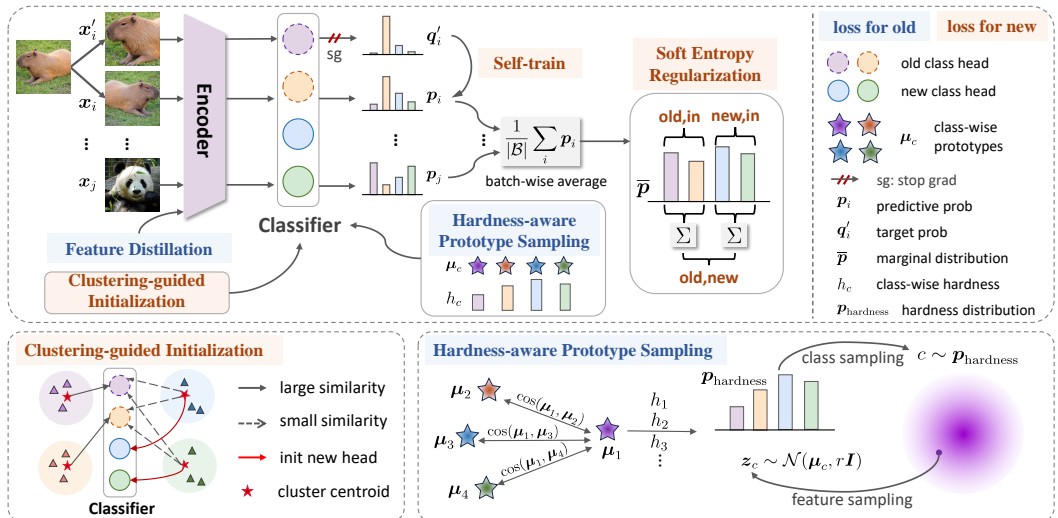

Figure 3: Illustration of the proposed Happy framework. **Top:** Overall learning pipeline for continual stages. **Bottom Left:** Clustering-guided Initialization, together with Soft Entropy Regularization (Section 4.2) ensures effective novel class category. **Bottom Right:** Hardness-aware Prototype Sampling (Section 4.3) remarkably mitigates catastrophic forgetting of old classes.

where $\mathcal{P}(i)$ is the positive set with the same label and $\tau_c$ is temperature. The overall loss function is:

$$\mathcal{L}_{\text{initial}} = \mathcal{L}_{\text{cls}} + \lambda_0 \mathcal{L}_{\text{con}}^{l} + (1 - \lambda_0) \mathcal{L}_{\text{con}}^{u}. \tag{2}$$

## 4.2 Classifier Initialization and Soft Entropy Regularization

Continuously discovering unlabeled new classes is challenging, as *prediction bias* towards old classes could collapse new class accuracy (Section 3.2). Therefore, we need to constrain the model to pay more attention to new classes to ensure effective category discovery.

**Clustering-guided Initialization.** Randomly initialized classifiers bring about unstable training. We argue that clustering could provide a good initialization for new classes. Specifically, at Stage-$t$, we employ KMeans [43] on $\mathcal{D}_{\text{train}}^{t}$ and obtain $K^t = K_{\text{old}}^t + K_{\text{new}}^t$ $\ell_2$-normalized cluster centroids $\{c_i\}_{i=1}^{K^t}$. Among them, the $K_{\text{new}}^t$ centroids least similar to old heads, as measured by maximum cosine similarity with them, serve as the potential initialization for new class heads:

$$\{t_j\}_{j=1}^{K_{\text{new}}^t} = \text{topk}_{t_j}(-\max_i c_{t_j}^\top \phi_i^{\text{old}}), \ \ i = 1, \cdots, K_{\text{old}}^t. \ \ \Rightarrow \ \ \phi_j^{\text{new}} = c_{t_j}, \ \ j = 1, \cdots, K_{\text{new}}^t. \tag{3}$$

**Group-wise Soft Entropy Regularization.** Entropy regularization [9, 29, 30] is common to avoid trivial solutions of clustering in *static* settings. However, at each stage of C-GCD, there are generally more old classes. Directly employing it equally across all classes will allocate most of the probability to old classes, leading to *prediction bias* and collapsed performance (as in Figure 2 (a, b)). To address this, we need to constrain the model explicitly. Considering that at each stage, there are fewer new classes but more samples per new class, and old classes have been well-learned previously, we propose to treat all old classes as a whole and the new classes as another, and derive C-GCD as binary classification. Specifically, we first compute the marginal probability in the batch $\overline{\boldsymbol{p}} \in \mathbb{R}^{K^t} = \frac{1}{|B|} \sum_{i \in B} \boldsymbol{p}_i$. Thus, $\overline{p}_{\text{old}} \in \mathbb{R} = \sum_{c \in \mathcal{C}_{\text{old}}^t} \overline{\boldsymbol{p}}^{(c)}$ and $\overline{p}_{\text{new}} \in \mathbb{R} = \sum_{c \in \mathcal{C}_{\text{new}}^t} \overline{\boldsymbol{p}}^{(c)}$ are scalars indicating the marginal distribution on old and new classes respectively, where the superscript $(c)$ denotes class indices and $\overline{p}_{\text{old}} + \overline{p}_{\text{new}} = 1$. Then we propose soft entropy regularization on the marginal distribution of the old and the new:

$$\mathcal{L}_{\text{entropy}}^{\text{old,new}} = \overline{p}_{\text{old}} \log \overline{p}_{\text{old}} + \overline{p}_{\text{new}} \log \overline{p}_{\text{new}}. \tag{4}$$

In this way, the model could focus more on each new class, ensuring reliable learning in new classes. We also employ entropy regularization within the new and old classes to avoid trivial solutions:

$$\mathcal{L}_{\text{entropy}}^{\text{old,in}} = \sum_{c \in \mathcal{C}_{\text{old}}^t} \overline{\boldsymbol{p}}^{(c)} \log \overline{\boldsymbol{p}}^{(c)}, \qquad \mathcal{L}_{\text{entropy}}^{\text{new,in}} = \sum_{c \in \mathcal{C}_{\text{new}}^t} \overline{\boldsymbol{p}}^{(c)} \log \overline{\boldsymbol{p}}^{(c)}. \tag{5}$$

To sum up, the soft entropy regularization is employed in a group-wise manner on three groups, *i.e.*, "inter old-new" (Eq. (4)), "intra old" and "intra new" (Eq. (5)), and we add them together:

$$\mathcal{L}_{\text{entropy-reg}} = \mathcal{L}_{\text{entropy}}^{\text{old,new}} + \mathcal{L}_{\text{entropy}}^{\text{old,in}} + \mathcal{L}_{\text{entropy}}^{\text{new,in}}. \tag{6}$$

The soft regularization ensures effective learning of new classes. See Section 5.4 for more discussions.

**Overall Loss for New Class Discovery.** To achieve self-training on unlabeled data, we perform self-distillation [9, 37]. Specifically, we use another augmented view $x_i'$ to produce sharpened $q_i'$ with smaller temperature $\tau_t < \tau_p$ and employ cross-entropy loss to supervise the prediction $p_i$: $\mathcal{L}_{\text{self-train}} = \frac{1}{2|B|} \sum_{i \in B} \ell(\mathbf{q}_i', \mathbf{p}_i) + \ell(\mathbf{q}_i, \mathbf{p}_i')$. The overall objective for new category discovery is:

$$\mathcal{L}_{\text{new}} = \mathcal{L}_{\text{self-train}} + \lambda_1 \mathcal{L}_{\text{entropy-reg}}, \tag{7}$$

where $\lambda_1$ controls the importance of the proposed regularization loss.

### 4.3 Hardness-aware Prototype Sampling

**Modeling Learned Classes.** Catastrophic forgetting [44, 45, 46] is a notorious problem in continual learning, especially when previous samples are inaccessible. Instead of storing seen samples, we can model the feature distribution for learned classes. Since the data in each incremental stage are unlabeled, at the end of each incremental stage, we perform class-wise Gaussian distribution in the feature space using models' predictions on $\mathcal{D}_{\text{train}}^t$:

$$\boldsymbol{\mu}_c = \frac{1}{N_c} \sum_{\hat{y}_i = c} \boldsymbol{f}_\theta^t(\boldsymbol{x}_i), \quad \boldsymbol{\Sigma}_c = \frac{1}{N_c} \sum_{\hat{y}_i = c} (\boldsymbol{f}_\theta^t(\boldsymbol{x}_i) - \boldsymbol{\mu}_c)(\boldsymbol{f}_\theta^t(\boldsymbol{x}_i) - \boldsymbol{\mu}_c)^\top, \quad c = 1, \cdots, K^t, \tag{8}$$

where $\hat{y}_i = \arg\max_c \boldsymbol{p}_i^{(c)}$ denotes the prediction, $\boldsymbol{\mu}_c$ and $\boldsymbol{\Sigma}_c$ are mean and covariance. Note that, for Stage-0, we directly use the ground-truth labels instead of predictions in Eq. (8). We call $\boldsymbol{\mu}_c$ as prototypes. When learning new knowledge, one can sample features from old classes $\mathcal{N}(\boldsymbol{\mu}_c, \boldsymbol{\Sigma}_c)$, and classify them correctly to mitigate forgetting. We find a shared diagonal matrix [47] empirically works fine, *i.e.*, $\boldsymbol{\Sigma}_c = r\boldsymbol{I}$, where $r$ is computed at Stage-0 as $r^2 = \frac{1}{K^0} \sum_{c \in \mathcal{C}_{\text{init}}^0} \text{Tr}(\boldsymbol{\Sigma}_c)/d$.

**Incorporating Hardness to Learned Classes.** As in Figure 2 (c), accuracy fluctuations across classes are significant, and treating all classes equally leads to *hardness bias* and suboptimal results. Intuitively, difficult classes should receive more attention during sampling. Here, we propose an unsupervised metric, considering the samples with higher similarity to others are more prone to be confused and therefore more difficult. We define hardness $h_i$ and obtain hardness distribution as:

$$h_i = \frac{1}{K_{\text{old}}^t - 1} \sum_{j=1, j \neq i}^{K_{\text{old}}^t} \cos(\boldsymbol{\mu}_i, \boldsymbol{\mu}_j) \quad \Rightarrow \quad \boldsymbol{p}_{\text{hardness}}^{(i)} = \sigma(h_i/\tau_h) = \frac{\exp(h_i/\tau_h)}{\sum_{j=1}^{K_{\text{old}}^t} \exp(h_j/\tau_h)}, \tag{9}$$

where $i = 1, \cdots, K_{\text{old}}^t$ and $\boldsymbol{p}_{\text{hardness}}$ is the categorical distribution to sample classes. Those with higher hardness are more likely to be sampled, which better suppresses the forgetting of hard classes.

**Sequential Sampling.** We first sample categories from categorical distribution $c \sim \boldsymbol{p}_{\text{hardness}}$ and then sample class-wise features from Gaussian distribution of the sampled classes $\boldsymbol{z}_c \sim \mathcal{N}(\boldsymbol{\mu}_c, r\boldsymbol{I})$ for classification. The loss for hardness-aware prototype sampling is :

$$\mathcal{L}_{\text{hap}} = \mathbb{E}_{c \sim p_{\text{hardness}}} \mathbb{E}_{\boldsymbol{z}_c \sim \mathcal{N}(\boldsymbol{\mu}_c, r\boldsymbol{I})} - y_c \log \sigma(\boldsymbol{g}_\phi^t(\boldsymbol{z}_c)/\tau_p). \tag{10}$$

**Overall Loss for Mitigating Forgetting.** As training proceeds, the feature space becomes outdated for previous prototypes, we thus apply knowledge distillation [48] using the last stage model and current training dataset, *i.e.*, $\mathcal{L}_{\text{kd}} = \frac{1}{|B|} \sum_{i \in B} 1 - \cos(\boldsymbol{f}_\theta^t(\boldsymbol{x}_i), \boldsymbol{f}_\theta^{t-1}(\boldsymbol{x}_i))$. The overall loss is:

$$\mathcal{L}_{\text{old}} = \mathcal{L}_{\text{hap}} + \lambda_2 \mathcal{L}_{\text{kd}}, \tag{11}$$

where $\lambda_2$ controls the weight of the knowedge distillation.

### 4.4 Overall Learning Objective

To continually discover new classes without forgetting old ones, we combine the losses for new (Eq. (7)) and old classes (Eq. (11)), and contrastive learning (Eq. (1)) to formulate the final objective:

$$\mathcal{L}_{\text{Happy}} = \mathcal{L}_{\text{new}} + \mathcal{L}_{\text{old}} + \lambda_3 \mathcal{L}_{\text{con}}^u. \tag{12}$$

Table 1: Performance of 5-stage Continual Generalized Category Discovery (C-GCD) on CIFAR100 (C100), ImageNet-100 (IN100), TinyImageNet (Tiny) and CUB. All methods have similar Stage-0 (S-0) ACC, which is fair for evaluation on continual stages. Here † denotes adjusted results.

| Datasets | Methods | S-0 | Stage-1 | | | Stage-2 | | | Stage-3 | | | Stage-4 | | | Stage-5 | | |
|---|---|---|---|---|---|---|---|---|---|---|---|---|---|---|---|---|---|
| | | All | All | Old | New | All | Old | New | All | Old | New | All | Old | New | All | Old | New |
| C100 | KMeans [43] | 66.16 | 40.27 | 41.76 | 32.80 | 37.14 | 38.33 | 30.00 | 36.20 | 37.63 | 26.20 | 36.66 | 38.30 | 23.50 | 35.69 | 36.79 | 25.80 |
| | VanillaGCD [7] | 90.82 | 72.32 | 78.50 | 41.40 | 67.04 | 72.50 | 34.30 | 57.99 | 62.26 | 28.10 | 56.60 | 59.55 | 33.00 | 51.36 | 53.70 | 30.30 |
| | SimGCD [9] | 90.36 | 73.37 | 86.44 | 8.00 | 62.56 | 72.43 | 3.30 | 54.17 | 61.61 | 2.10 | 47.62 | 53.37 | 1.60 | 43.53 | 47.86 | 4.60 |
| | SimGCD+ [44] | 90.36 | 75.93 | **87.04** | 20.40 | 67.07 | 75.33 | 17.50 | 58.45 | 64.33 | 17.30 | 54.31 | 58.71 | 19.10 | 50.49 | 53.90 | 19.80 |
| | FRoST [12] | 90.36 | 76.87 | 79.58 | **63.30** | 65.31 | 68.88 | 43.90 | 58.01 | 61.09 | 36.50 | 49.27 | 50.90 | 36.20 | 48.03 | 48.17 | 46.80 |
| | GM [15]† | 90.36 | 76.58 | 79.80 | 60.50 | 71.10 | 74.52 | **50.60** | 63.51 | 68.16 | 31.00 | 59.74 | 62.51 | 37.60 | 54.11 | 54.74 | 48.40 |
| | MetaGCD [18] | 90.82 | 76.12 | 83.60 | 38.70 | 69.40 | 72.82 | 48.90 | 61.95 | 65.76 | 35.30 | 58.22 | 61.21 | 34.30 | 55.78 | 58.47 | 31.60 |
| | Happy (Ours) | 90.36 | **80.40** | 85.26 | 56.10 | **74.13** | 78.27 | 49.30 | **68.23** | 70.86 | 49.80 | **62.26** | 63.75 | 50.30 | **59.99** | 60.96 | 51.30 |
| IN100 | KMeans [43] | 85.56 | 54.90 | 57.04 | 44.20 | 54.73 | 56.37 | 44.90 | 54.67 | 56.66 | 40.80 | 54.63 | 56.25 | 41.70 | 53.92 | 56.18 | 33.60 |
| | VanillaGCD [7] | 95.96 | 70.13 | 72.92 | 56.20 | 69.37 | 73.47 | 44.80 | 68.50 | 70.63 | 53.60 | 65.56 | 67.85 | 47.20 | 64.54 | 67.44 | 38.40 |
| | SimGCD [9] | 96.20 | 79.67 | 91.68 | 19.60 | 70.23 | 78.83 | 18.60 | 61.90 | 67.43 | 23.20 | 56.67 | 60.92 | 22.60 | 52.90 | 56.40 | 21.40 |
| | SimGCD+ [44] | 96.20 | 83.07 | 95.16 | 22.60 | 74.57 | 83.47 | 21.20 | 67.60 | 73.57 | 25.80 | 62.09 | 66.83 | 24.20 | 57.62 | 61.47 | 23.00 |
| | FRoST [12] | 96.20 | 87.50 | 92.96 | 60.20 | 79.63 | 83.37 | 57.20 | 76.78 | 77.00 | 75.20 | 66.18 | 68.65 | 46.40 | 63.82 | 66.40 | 40.60 |
| | GM [15]† | 96.20 | 89.53 | 95.04 | 62.00 | 82.34 | 86.93 | 54.80 | 77.97 | 79.17 | 69.60 | 72.80 | 74.65 | 58.00 | 71.08 | 71.76 | 65.00 |
| | MetaGCD [18] | 95.96 | 75.27 | 78.20 | 60.60 | 73.79 | 75.93 | 54.90 | 69.35 | 72.20 | 49.40 | 67.22 | 70.10 | 44.20 | 66.68 | 69.31 | 43.00 |
| | Happy (Ours) | 96.20 | **91.20** | 95.36 | 70.40 | **87.83** | 90.83 | 69.80 | **85.22** | 86.40 | 77.00 | **81.93** | 83.00 | 73.40 | **78.58** | 79.11 | 73.80 |
| Tiny | KMeans [43] | 61.70 | 35.42 | 35.46 | 35.20 | 34.99 | 35.75 | 30.40 | 34.80 | 36.07 | 25.90 | 34.77 | 35.90 | 24.90 | 34.62 | 35.63 | 25.50 |
| | VanillaGCD [7] | 84.20 | 55.93 | 58.92 | 41.00 | 54.96 | 58.58 | 33.20 | 52.82 | 55.74 | 32.40 | 48.81 | 51.46 | 27.60 | 45.94 | 48.06 | 26.90 |
| | SimGCD [9] | 85.86 | 66.95 | 79.94 | 2.00 | 57.81 | 66.98 | 2.80 | 52.70 | 59.83 | 2.77 | 45.01 | 50.29 | 2.80 | 41.59 | 45.79 | 3.80 |
| | SimGCD+ [44] | 85.86 | 70.38 | 81.80 | 13.30 | 62.47 | 70.75 | 12.80 | 54.55 | 60.46 | 13.20 | 47.98 | 52.49 | 11.90 | 42.98 | 46.46 | 12.70 |
| | FRoST [12] | 85.86 | 75.15 | 78.56 | 58.10 | 65.64 | 67.83 | **52.50** | 51.32 | 54.31 | 30.40 | 48.22 | 52.14 | 16.90 | 40.15 | 42.73 | 16.90 |
| | GM [15]† | 85.86 | 76.42 | **82.40** | 46.50 | 68.87 | 73.82 | 39.20 | 58.68 | 63.43 | 25.40 | 52.86 | 57.21 | 18.10 | 46.90 | 50.62 | 13.40 |
| | MetaGCD [18] | 84.20 | 60.88 | 64.90 | 40.80 | 57.20 | 61.03 | 34.20 | 54.36 | 57.19 | 34.60 | 50.83 | 53.59 | 28.80 | 48.14 | 50.16 | 30.00 |
| | Happy (Ours) | 85.86 | **78.85** | 82.40 | 61.10 | **71.34** | 76.18 | 42.30 | **64.68** | 68.70 | 36.50 | **58.49** | 60.64 | 41.30 | **54.56** | 56.66 | 35.70 |
| CUB | KMeans [43] | 43.93 | 32.54 | 30.76 | 41.18 | 31.19 | 30.53 | 35.20 | 29.28 | 27.46 | 42.09 | 29.19 | 28.13 | 37.61 | 28.17 | 27.01 | 38.53 |
| | VanillaGCD [7] | 89.20 | 64.47 | 67.06 | 51.93 | 58.15 | 60.65 | 42.91 | 54.10 | 56.40 | 37.91 | 49.98 | 51.33 | 39.32 | 46.84 | 46.58 | 49.14 |
| | SimGCD [9] | 90.26 | 73.84 | 84.54 | 22.02 | 63.36 | 72.35 | 8.58 | 55.63 | 61.95 | 11.13 | 49.31 | 54.55 | 7.86 | 44.72 | 48.69 | 9.25 |
| | SimGCD+ [44] | 90.26 | 75.62 | **85.55** | 25.97 | 65.32 | 73.93 | 13.68 | 57.40 | 63.28 | 16.26 | 51.11 | 55.72 | 14.27 | 45.79 | 49.29 | 14.28 |
| | FRoST [12] | 90.26 | 77.03 | 83.95 | 43.53 | 50.77 | 53.46 | 34.33 | 46.42 | 49.31 | 26.09 | 39.40 | 41.47 | 23.08 | 34.55 | 35.12 | 29.45 |
| | GM [15]† | 90.26 | 76.17 | 80.23 | 56.51 | 67.91 | 73.38 | 34.58 | 61.12 | 66.53 | 23.00 | 55.90 | 57.49 | 43.38 | 51.96 | 54.40 | 30.10 |
| | MetaGCD [18] | 89.20 | 67.08 | 70.21 | 51.92 | 60.77 | 62.39 | 50.86 | 57.53 | 59.33 | 37.78 | 51.90 | 52.22 | 49.40 | 49.60 | 49.96 | 46.38 |
| | Happy (Ours) | 90.26 | **81.40** | 85.06 | 63.70 | **74.27** | 76.03 | 63.57 | **67.09** | 71.06 | 39.13 | **62.25** | 63.83 | 49.74 | **59.39** | 60.49 | 49.52 |

## 5 Experiments

### 5.1 Experimental Setup

**Datasets.** We construct C-GCD on four datasets: CIFAR100 [36] (C100), ImageNet-100 [49] (IN100), Tiny-ImageNet [50] (Tiny) and CUB [51], each is split into two subsets: (1) Stage-0, where 50% of classes serving as $\mathcal{C}_{\text{init}}^0$ constitute initial **labeled** data. (2) Stage-$1 \sim T$ ($T = 5$ by default). At each stage, the remaining classes are evenly sampled as new classes, along with all previously learned classes to constitute continual **unlabeled** data. Detailed dataset statistics are shown in Table 2.

Table 2: Dataset splits of C-GCD setting. We show #classes and #images *per class* of different stages. #old denotes all previously learned classes.

| Datatset | Stage-0 | | Each Stage-t ($t = 1, \cdots, 5$) | | |
|---|---|---|---|---|---|
| | #class | #img/#class | #new | #img/#new | #img/#old |
| C100 | 50 | 400 | 10 | 400 | 25 |
| IN100 | 50 | ~1,000 (80%) | 10 | 1,000 | 60 |
| Tiny | 100 | 400 | 20 | 400 | 25 |
| CUB | 100 | ~25 (80%) | 20 | 25 | 5 |

**Evaluation Protocol.** At each stage, after training on $\mathcal{D}_{\text{train}}^t$, the model is evaluated on disjoint test $\mathcal{D}_{\text{test}}^t$, *i.e.*, *inductive* setting, which contains both new $\mathcal{C}_{\text{new}}^t$ and old classes $\mathcal{C}_{\text{old}}^t$. The accuracy is calculated using ground truth $y_i$ and models' predictions $\hat{y}_i$ as: $ACC = \max_{p \in \mathcal{P}(\mathcal{C}_t)} \frac{1}{M} \sum_{i=1}^{M} \mathbb{1}(y_i = p(\hat{y}_i))$, where $M = |\mathcal{D}_{\text{test}}^t|$ and $\mathcal{P}(\mathcal{C}^t)$ is the set of all permutations across all classes $\mathcal{C}_{\text{old}}^t \cup \mathcal{C}_{\text{new}}^t$. The optimal permutation could be computed *once* using Hungarian algorithm [52] on all classes, and we report "All", "Old" and "New" accuracies as evaluate metrics.

**Implementation Details.** Following the convention [7, 10, 18, 31], we use ViT-B/16 [53] pretrained by DINO [37] as the backbone, and fine-tune only the last transformer block for all experiments. The output [CLS] token is chosen as feature representation. At Stage-0, models are trained

Table 3: Forgetting & discovery.

| Methods | C100 | | Tiny | |
|---|---|---|---|---|
| | $\mathcal{M}_f \downarrow$ | $\mathcal{M}_d \uparrow$ | $\mathcal{M}_f \downarrow$ | $\mathcal{M}_d \uparrow$ |
| VanillaGCD | 17.10 | 33.42 | 20.20 | 32.22 |
| FRoST | 22.82 | 45.34 | 21.62 | 34.96 |
| MetaGCD | 16.56 | 37.76 | 19.30 | 33.68 |
| Happy | **11.22** | **51.36** | **9.75** | **43.38** |

Table 4: 'All' ACC of C-GCD across 10 continual stages.

| Data | Methods | 0 | 1 | 2 | 3 | 4 | 5 | 6 | 7 | 8 | 9 | 10 |
|---|---|---|---|---|---|---|---|---|---|---|---|---|
| | VanillaGCD | 90.82 | 78.42 | 75.68 | 70.35 | 66.64 | 64.29 | 61.05 | 58.33 | 57.14 | 56.23 | 55.15 |
| C100 | MetaGCD | 90.82 | 81.07 | 76.55 | 74.26 | 67.64 | 64.45 | 61.58 | 59.13 | 60.13 | 56.91 | 56.51 |
| | Happy | 90.36 | **85.62** | **81.88** | **79.82** | **74.01** | **71.81** | **68.46** | **64.05** | **62.14** | **61.38** | **57.81** |
| | VanillaGCD | 84.20 | 65.15 | 64.63 | 60.94 | 59.46 | 56.52 | 55.47 | 51.65 | 50.66 | 49.83 | 48.56 |
| Tiny | MetaGCD | 84.20 | 68.87 | 65.48 | 62.92 | 60.81 | 58.21 | 56.16 | 54.68 | 52.58 | 50.57 | 48.92 |
| | Happy | 85.86 | **80.75** | **76.92** | **73.34** | **69.77** | **66.33** | **62.75** | **57.56** | **54.73** | **53.02** | **50.69** |

Table 5: Ablations on the main components. Average accuracies of 5 stages are reported.

| ID | Category Discovery | | Mitigating Forgetting | | CIFAR100 | | | CUB | | |
|---|---|---|---|---|---|---|---|---|---|---|
| | $\mathcal{L}_{\text{entropy-reg}}$ | init | $\mathcal{L}_{\text{hap}}$ | $\mathcal{L}_{\text{kd}}$ | All | Old | New | All | Old | New |
| (a) | ✗ | ✗ | ✗ | ✗ | 50.95 | 58.66 | 1.96 | 53.28 | 60.75 | 4.70 |
| (b) | ✓ | ✗ | ✗ | ✗ | 57.67 | 65.58 | 7.44 | 59.26 | 65.99 | 14.91 |
| (c) | ✗ | ✓ | ✗ | ✗ | 58.26 | 65.33 | 12.84 | 63.11 | 68.62 | 27.33 |
| (d) | ✓ | ✓ | ✗ | ✗ | 60.51 | 67.39 | 16.36 | 64.53 | 69.34 | 32.91 |
| (e) | ✗ | ✗ | ✓ | ✓ | 57.75 | 66.32 | 1.96 | 57.67 | 66.05 | 3.69 |
| (f) | ✓ | ✓ | ✓ | ✗ | 66.89 | 69.98 | 47.94 | 66.36 | 70.94 | 37.15 |
| (g) | ✓ | ✓ | ✓ | ✓ | **69.00** | **71.82** | **51.36** | **68.88** | **71.29** | **53.13** |

with 100 epochs. Subsequently, we train models 30 epochs at each continual stage with a batch size of 128 and a learning rate of 0.01. We set $\{\lambda_1, \lambda_2, \lambda_3\}$ as 1 and temperature $\{\tau_p, \tau_h\}$ as 0.1 while $\tau_t$ as 0.05. All experiments are run on NVIDIA GeForce RTX 4090 GPUs.

## 5.2 Comparison with State-of-the-Arts

We compare our methods with (1) Kmeans [43] on pre-trained features, (2) GCD methods: VanillaGCD [7], SimGCD [9], SimGCD+LwF [44], and (3) recent continual category discovery works: FRoST [12], GM [15] and MetaGCD [18]. Since GM [15] requires storing exemplar samples, we adjust it to sampling features. For a fair comparison, all methods use the same objective (Eq. (2)) to pre-train the model at Stage-0. Results are reported in Table 1, Table 3, and Table 4.

Happy **outperforms prior methods by a large margin.** For example in Table 1, on IN100, compared to MetaGCD [18] and GM [15], our approach achieves an improvement of 11.90% and 7.50% for 'All' accuracy, respectively. On C100, Happy improves the previous state-of-the-art by 3.45% and 13.60% for old and new classes across 5 stages. Besides, our method produces more balanced accuracy between 'Old' and 'New'. These improvements benefit from our consideration of underlying bias in the task of C-GCD and the tailor-made debiased components in Happy.

Happy **effectively balances discovering new classes with mitigating forgetting old classes.** To decouple and analyze the two conflicting objectives, we use $\mathcal{M}_f$ and $\mathcal{M}_d$ in [15] to evaluate the overall forgetting of labeled classes and the discovery of new classes respectively. Table 3 shows that VanillaGCD [7] and MetaGCD [18] struggle with category discovery due to the weak supervision of contrastive learning. In addition, FRoST [12] focuses solely on new classes, at the expense of old class performance. In contrast, our method effectively balances both, achieving improvements of 6~12% in two metrics.

**C-GCD with more continual stages.** To explore more realistic and challenging scenarios, we conduct C-GCD with 10 continual stages. Results in Table 4 demonstrate that Happy consistently outperforms other counterparts, showcasing Happy is a competent long-term novel category discoverer.

## 5.3 Ablation Study

Here, we conduct extensive ablations on each main component (Table 5) and analyze how our method handles the conflicting goals between discovering new classes and mitigating forgetting old ones. Finally, we delve into the mechanism of hardness in our framework.

**How does Happy achieve remarkable category discovery?** In Table 5 (a), we observe that models trained with only $\mathcal{L}_{\text{self-train}}$ are collapsed in 'New' ACC. (b) and (c) incorporate soft entropy regularization and the designed initialization, respectively. In addition, (d) combines both of them and brings significant improvements for new classes, *e.g.*, 28.21% on CUB. From (a) to (d), the initialization

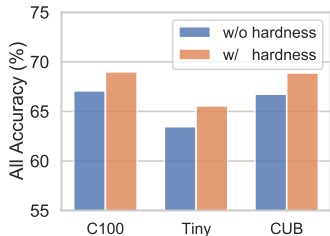

Figure 4: Effect of hardness.

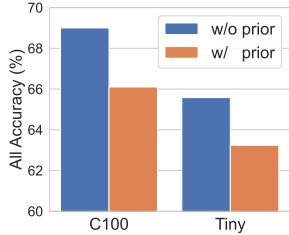

Figure 5: Analysis: Acc.

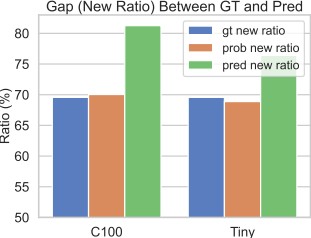

Figure 6: Analysis: Ratio.

produces robust and desirable feature location, and $\mathcal{L}_{\text{entropy-reg}}$ mitigates *prediction bias* and ensures necessary learning of new classes. Additionally, mitigating the forgetting of old classes also helps ((d)→(g)), as it ensures the preservation of general representations for most classes, which in turn benefits the clustering of new classes.

**How does** `Happy` **mitigate catastrophic forgetting?** (f) includes hardness-aware sampling based on (d), which improves 'Old' ACC by 2.59% on CIFAR100. However, without $\mathcal{L}_{\text{kd}}$, the feature space could drift significantly when learning new classes and become misaligned with the learned classifier, which degrades the performance. As a whole, (g) incorporates $\mathcal{L}_{\text{kd}}$ to remarkably improve 'Old' by 4.43% and 1.95% on CIFAR100 and CUB. Similarly, better clustering of new classes also benefits old ones because incorrectly classifying new as old can hinder the learning of old classes. In this sense, the learning of new and old classes is mutually reinforcing.

**How does hardness-awareness help C-GCD?** To delve into the effectiveness of hardness-aware modeling, we conduct ablations with and without it. Results (average 'All' accuracy across 5 stages) in Figure 4 show that hardness-awareness consistently improves performance across various datasets. We also present sensitivity analysis on temperature $\tau_h$ in Eq. (9). As Table 6 shows, $\tau_h = 0.1$ is a proper choice. When $\tau_h$ is too large, $p_{\text{hardness}}$ convergences to the uniform distribution, which is similar to the one without hardness modeling. A small $\tau_h$ also brings suboptimal results. In such cases, $p_{\text{hardness}}$ becomes overly sharp, resulting in the sampling of only a very limited number of hard classes, which exacerbates the forgetting of remaining categories.

Table 6: Sensitivity analysis of $\tau_h$.

| $\tau_h$ | C100 | Tiny |
|---|---|---|
| 0.01 | 66.40 | 62.36 |
| 0.05 | 68.06 | 64.95 |
| 0.1 | **69.00** | **65.56** |
| 1 | 68.01 | 64.76 |
| 10 | 67.59 | 63.93 |

### 5.4 Further Analysis

**Does incorporating class prior into regularization necessarily improve results?** Without any prior knowledge about the proportion of new and old class samples, we employ soft entropy regularization in Eq. (4) to prevent bias. A natural question arises: *Can the introduction of information about the ratio of new to old class samples at each stage further enhance performance?* To explore this issue, we directly use the ground truth ratio of old and new samples $\overline{p}_{\text{old}}^{\text{gt}}, \overline{p}_{\text{new}}^{\text{gt}}$ as a prior and modify Eq. (4) as $\mathcal{L}_{\text{prior}}^{\text{old,new}} = -\overline{p}_{\text{old}}^{\text{gt}} \log \overline{p}_{\text{old}} - \overline{p}_{\text{new}}^{\text{gt}} \log \overline{p}_{\text{new}}$. That is, using cross-entropy to supervise the model's predictive probabilities $\overline{p}_{\text{old}}, \overline{p}_{\text{new}}$, which surprisingly degrades performance as shown in Figure 5. The reason lies in the gap between the model's predicted ratio of new class samples (`pred new ratio`) and the prior ratio of new classes $\overline{p}_{\text{new}}^{\text{gt}}$ (`gt ratio`), as revealed in Figure 6, which is caused by the confidence gap between old and new classes (Figure 2). This gap ultimately causes the predicted ratio of new samples to exceed $\overline{p}_{\text{new}}^{\text{gt}}$, bringing about degraded performance than using Eq. (4) without any prior.

**Unknown class number scenarios.** Previous experiments assume the number of new classes $K_{\text{new}}^t$ is known, which often does not hold in reality. At the start of each stage, we need to first estimate the number of new classes before instantiating the classifier. Prior arts [5, 7] query some labeled data when estimating the class number, which is not applicable in the purely unsupervised setting of C-GCD. Instead, we employ off-the-shelf *silhouette score* [35] to estimate $K_{\text{new}}^t$ in an unsupervised manner. Specifically, we compute *silhouette score* using mean intra-cluster distance and mean nearest-cluster distance

Table 7: Unknown class number results on C100.

| Methods | All | Old | New |
|---|---|---|---|
| GCD | 58.72 | 62.66 | 32.92 |
| MetaGCD | 63.28 | 67.65 | 34.94 |
| Ours | **68.80** | **72.40** | **45.74** |

Table 8: Effectiveness of proposed $\mathcal{L}_{\text{entropy-reg}}$ and hardness-aware modeling for bias mitigation.

|  | CIFAR100 | | CUB | |
|---|---|---|---|---|
|  | $\Delta p \downarrow$ | $\Delta r \downarrow$ | $\Delta p \downarrow$ | $\Delta r \downarrow$ |
| w/o $\mathcal{L}_{\text{entropy-reg}}$ | 81.50 | 63.25 | 83.20 | 65.80 |
| w/ $\mathcal{L}_{\text{entropy-reg}}$ | **5.76** | **10.20** | **10.25** | **11.05** |

(a) Mitigation of *probability bias*.

|  | CIFAR100 | | CUB | |
|---|---|---|---|---|
|  | $Var_0 \downarrow$ | $Acc_h \uparrow$ | $Var_0 \downarrow$ | $Acc_h \uparrow$ |
| w/o hardness | 23.04 | 65.10 | 21.77 | 62.65 |
| w/ hardness | **10.33** | **70.23** | **9.28** | **68.40** |

(b) Mitigation of *hardness bias*.

and select the number of classes corresponding to the highest score value as the estimation. Then we utilize the estimated number for training and evaluation. Average accuracies across 5 stages on CIFAR100 are reported in Table 7. Our method outperforms others when $K_{\text{new}}^t$ is not known *a-prior*.

`Happy` **could effectively mitigate two types of bias in C-GCD.** As elaborated in Section 3.2, models in C-GCD are susceptible to *prediction bias* and *hardness bias*. To validate the effectiveness of the proposed method in bias mitigation, we design metrics to quantitatively measure these biases. Specifically, for *prediction bias*, we provide two metrics: (1) $\Delta p(\downarrow) = \bar{p}_{\text{old}} - \bar{p}_{\text{new}}$ denotes the difference in marginal probabilities between old and new classes (see Section 4.2). (2) $\Delta r(\downarrow)$ denotes the proportion of new classes' samples misclassified as old classes. Both $\Delta p$ and $\Delta r$ are calculated on the test data after Stage-1. The results in Table 8a from two datasets demonstrate that $\mathcal{L}_{\text{entropy-reg}}$ effectively reduces prediction bias, with a significantly lower marginal probability gap and fewer new class samples misclassified as the old. For *hardness bias*, we also present two metrics: (1) $Var_0(\downarrow)$ denotes the variance in accuracy of the initial labeled classes $\mathcal{C}_{\text{init}}^0$. (2) $Acc_h(\uparrow)$ denotes the accuracy of the hardest class in $\mathcal{C}_{\text{init}}^0$. Both metrics are calculated after 5 stages. Results in Table 8b demonstrate that hardness-aware sampling effectively reduces *hardness bias*, with lower accuracy variance and higher hardest accuracy. In this regard, the proposed modules competently alleviate both types of bias, which is consistent with our motivation.

# 6 Conclusive Remarks

We tackle the pragmatic but underexplored task of Continual Generalized Category Discovery (C-GCD), which involves conflicting goals of continually discovering unlabeled new classes while preventing forgetting old ones. We further identify *prediction bias* and *hardness bias* hinder the effective learning of both old and new classes. To overcome these issues, we propose a debiased framework namely `Happy`. The clustering-guided initialization and soft entropy regularization collectively alleviate *prediction bias* and ensure the clustering of new classes. On the other hand, by modeling the hardness of learned classes, we propose hardness-aware prototype sampling to dynamically place more attention on difficult classes, which significantly prevents the forgetting of old classes. Overall, our method achieves better discovery of new classes with minimal forgetting of old classes, which is validated by extensive experiments across various scenarios.

**Limitations and Future Works.** Due to the imbalanced labeling conditions between the initial and continual stages in C-GCD, the model's confidence is not calibrated and there is an obvious confidence gap between old and new classes, in these cases, incorporating prior information even degrades performance (Section 5.4). Future work should incorporate confidence calibration [38] into C-GCD to further mitigate potential biases. Another promising direction is to devise competent class number estimation methods for C-GCD, because in the unsupervised setting, class number estimation becomes significantly challenging. Additionally, this paper primarily discusses classification tasks, while future works could extend the C-GCD learning paradigm to object detection [54], segmentation [55] and multi-modal learning [56, 57, 58].

## Acknowledgments and Disclosure of Funding

This work has been supported by the National Science and Technology Major Project (2022ZD0116500), National Natural Science Foundation of China (U20A20223, 62222609, 62076236), CAS Project for Young Scientists in Basic Research (YSBR-083), Key Research Program of Frontier Sciences of CAS (ZDBS-LY-7004), and the InnoHK program.

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

# A  More Discussions about the Task of C-GCD

In this section, we first provide a detailed explanation of the task of Continual Generalized Category Discovery (C-GCD) and a comparison with class-incremental learning. Then we illustrate the practicality of C-GCD studied in this paper through some examples.

## A.1  Comparison with Class-incremental Learning

The core differences between C-GCD and Class-incremental Learning [20, 44] (CIL) lie in that training data is fully unlabeled at each stage of C-CGD, by contrast, conventional CIL adopts a fully-supervised setting. On the other hand, at each stage of C-GCD, the unlabeled training data $\mathcal{D}_{\text{train}}^t$ contains samples from previously seen classes, which makes the task more challenging because models need to implicitly or explicitly split the samples from old and new classes and then discover novel categories. While in rehearsal-free CIL, at each stage, the labeled training dataset typically does not contain samples of previous classes, otherwise it becomes the replay-based sitting and will simplify the problem, because the training data is fully labeled.

## A.2  Realistic Considerations of C-GCD

As mentioned in the main manuscript, we study a more pragmatic setting of C-GCD, whose specific manifestations of realistic considerations are listed as follows:

**More continual stages with more novel categories to be discovered.**   Prior works [15, 16, 18] mainly implement C-GCD with 3 stages given nearly 70% of all the classes serving as labeled classes. This simple setting does not reflect real-world scenarios. Humans are lifelong learners over the course of their entire lives, and our setting closely aligns with this situation. Specifically, the default setting in this paper has 5 continual stages with 50% of all the classes serving as novel classes.

**Rehearsal-free setting without storing previous samples.**   Several works [15, 17] in C-GCD require the storage of previous samples to construct a non-parametric classifier or mitigate catastrophic forgetting. This store-and-replay manner could cause privacy and storage issues, especially in cases with very long learning periods. While we study the rehearsal-free C-CGD.

**The ratio of new class samples is unknown.**   Some works study C-GCD by assuming that the proportion of new class samples per stage is known, which facilitates the design of novelty detection, owing to the fact that novelty detection [15, 39] typically relies on a threshold to determine whether a sample is from novel classes. In our setting, we lift this restrictive assumption and our framework Happy does not rely on the ratio. Instead, our method does not explicitly perform novelty detection, but instead implicitly learns with soft entropy regularization and self-distillation.

**At each stage, the number of samples of each old class is significantly fewer than the number of samples in each new one.**   If at each stage, the number of class-wise samples of old and new classes is roughly the same or both have plenty of samples, then C-GCD degenerates to the static setting of GCD where the catastrophic forgetting is inherently avoided because there are plenty of samples for each class, which bring about desirable outcomes even using baseline methods. This also contradicts the reality. Imagine a scenario where a student is growing up and entering different stages of learning. For example, he is currently in college where he needs to self-learn many new subjects like *calculus* and *linear algebra*. However, he occasionally encounters some old knowledge from his high school days, such as *trigonometry* and *plane geometry*. In this case, new knowledge is mixed with old knowledge, but the quantity of old knowledge is quite small.

# B  More Implementation Details

**Fair training in Stage-0.**   We train all the methods using similar objectives at Stage-0, specifically, for methods with parametric classifiers [9, 12, 15], we employ $\mathcal{L}_{\text{init}}$ in Eq. (2), while for methods with contrastive learning and non-parametric classifiers [7, 18], we employ supervised and self-supervised contrastive learning on the labeled data, *i.e.*, the last two terms in Eq. (2). The results of different methods at Stage-0 are similar, as shown in Table 1, ensuring fair comparisons of subsequent continual learning stages.

**Model Details.** Following the convention of the literature [7, 10, 18], we use ViT-B/16 [53] pre-trained with DINO [37] as the encoder, and fine-tune only the last transformer block for all experiments. The output [CLS] token is chosen as feature representation. For the parametric classifier, we use $\ell_2$ weight normed prototypical classifier [9, 30] without the bias term. The dimensionality of feature space and projection space for contrastive learning is 768 and 65,536, respectively. Note that all the feature vectors in the 768-dimensional feature space are $\ell_2$-normalized, *i.e.*, hyperspherical feature space, including the feature representation $z_i$ of each sample $x_i$, the head of each class in the classifier $\phi_i$, the KMeans [43] cluster centroids $c_i$ in Eq. (3), the class-wise prototypes $\mu_c$ in Eq. (8) and the sampled features $z_c$ in Eq. (10).

**Training Details.** We train the models in Stage-0 for 100 epochs with a learning rate of 0.1, and 30 epochs with a learning rate of 0.01 for each of the continual stages. We use a cosine annealed schedule for the learning rate.

**Hyper-parameters and implementation details of** Happy**.** For the weights of loss terms, we empirically set $\lambda_0 = 0.35$ and $\lambda_1 = \lambda_2 = 1$, and detailed hyper-parameter analysis is elaborate in Section E.5. For the temperature, we set the main temperature $\tau_p = 0.1$ in model predictive probability $p_i$ and the $\tau_t$ in the sharp soft $q_i$. We set $\tau_h$ in hardness distribution $p_{\text{hardness}}$ as 1. As for the temperature in the contrastive learning term, we follow prior arts [7, 9] and set $\tau_c$ as 0.07 and 1 for supervised and self-supervised contrastive learning, respectively. When we compute $\mathcal{L}_{\text{entropy}}^{\text{old,in}}$ and $\mathcal{L}_{\text{entropy}}^{\text{new,in}}$ in Eq. (5), the distribution within old $\overline{p}^{(c)}, c \in \mathcal{C}_{\text{old}}^t$ and new classes $\overline{p}^{(c)}, c \in \mathcal{C}_{\text{new}}^t$ should be firstly normalized whose summation across the class indices equals to 1.

## C   Algorithm of the Proposed Method Happy

In this section, we give a detailed algorithm of Happy in Algorithm 1, including both (1) Initial supervised learning (Stage-0) and (2) Continual unsupervised discovery (Stage-1 $\sim T$).

---

**Algorithm 1** Training Pipeline of Happy

---

**Input:** Initial labeled dataset $\mathcal{D}_{\text{train}}^0 = \{(x_i^l, y_i)\}_{i=1}^{N^0}$ of $K^0$ classes $\mathcal{C}_{\text{init}}^0$, and training epochs $E_0$ for Stage-0.
**Input:** Number of continual stages $T$.
**Input:** Continual stages dataset $\{\mathcal{D}_{\text{train}}^t\}_{t=1}^T$ of classes $\mathcal{C}^t = \mathcal{C}_{\text{old}}^t \cup \mathcal{C}_{\text{new}}^t$ and training epochs $E$ for each stage.
**Input:** Number of new classes $K_{\text{new}}^t$ at each stage, which could be ground truth or estimated.
**Input:** The model $h^t = g_\phi^t \circ f_\theta^t(\cdot)$ where $f_\theta^t(\cdot)$ is encoder and $g_\phi^t$ is parametric classifier.
1: # ================================= Stage-0 =================================
2: **for** epoch $e = 1 \rightarrow E_0$ **do**
3:     Train the model $h^0$ on $\mathcal{D}_{\text{train}}^0$ using the loss function $\mathcal{L}_{\text{initial}}$ in Eq. (2).
4: **end for**
5: ▷ Compute class-wise prototypes $\mu_c$ $(c = 1, \cdots, K^0)$ in Eq. (8) using ground truth labels
6: ▷ Compute class-shared radius $r^2 = \frac{1}{K^0} \sum_{c \in \mathcal{C}_{\text{init}}^0} \text{Tr}(\Sigma_c)/d$
7: ▷ Model hardness distribution $p_{\text{hardness}}$ on all the prototypes using Eq. (9)
8:
9: # ============================= Continual Stages =============================
10: **for** epoch $t = 1 \rightarrow T$ **do**
11:     ▷ Clustering-guided initialization of current new heads $\{\phi_j^{\text{new}}\}_{j=1}^{K_{\text{new}}^t}$ using Eq. (3)
12:     **for** epoch $e_t = 1 \rightarrow E$ **do**
13:         ▷ Train the model $h^t$ on $\mathcal{D}_{\text{train}}^t$ using the overall loss function $\mathcal{L}_{\text{Happy}}$ in Eq. (12)
14:     **end for**
15:     ▷ Compute class-wise prototypes $\mu_c$ $(c = K^{t-1}+1, \cdots, K^t)$ in Eq. (8) using model predictions
16:     ▷ Append new prototypes to the existing set
17:     ▷ Model hardness distribution $p_{\text{hardness}}$ on all the prototypes using Eq. (9)
18: **end for**
**Output:** The trained model $h^T = g_\phi^T \circ f_\theta^T(\cdot)$ that could perform classification on all seen classes.

---

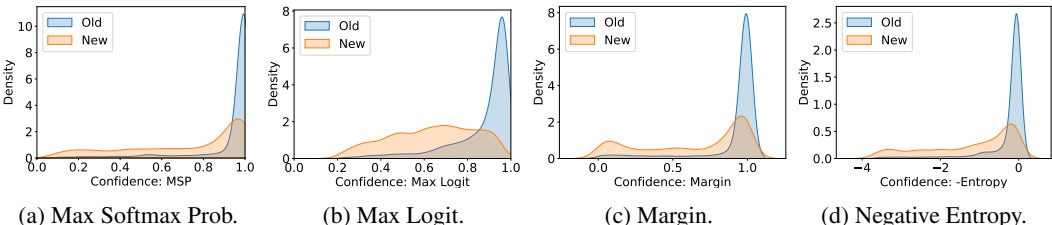

| (a) Max Softmax Prob. | (b) Max Logit. | (c) Margin. | (d) Negative Entropy. |

Figure 7: Confidence gap of various metrics between old and new classes of baseline models.

# D    Metrics of C-GCD

C-GCD is essentially a clustering problem, specifically for the unlabeled new classes. Following [7, 5, 9, 10, 18], the accuracy is calculated using ground truth $y_i$ and models' predictions $\hat{y}_i$

$$ACC = \max_{p \in \mathcal{P}(\mathcal{C}^t)} \frac{1}{M} \sum_{i=1}^{M} \mathbb{1}(y_i = p(\hat{y}_i)), \tag{13}$$

here, $M = |\mathcal{D}_{\text{test}}^t|$ is the number of samples in the test dataset and $\mathcal{P}(\mathcal{C}^t)$ represents the set of all permutations across all classes $\mathcal{C}_{\text{old}}^t \cup \mathcal{C}_{\text{new}}^t$. The optimal permutation could be computed *once* using Hungarian algorithm [52], and subsequently 'All', 'Old' and 'New' are computed on corresponding indices of classes. C-GCD utilizes *inductive* evaluation, *i.e.*, models are evaluated on a disjoint test dataset containing all of the seen classes.

To decouple and analyze the objectives of novel class discovery and preventing forgetting, GM [15] designed new metrics, *i.e.*, the maximum forgetting metric $\mathcal{M}_f$ and the final discovery metric $\mathcal{M}_d$. They are defined as follows:

$$\mathcal{M}_f = \max_t \{ACC_{\text{old}}^0 - ACC_{\text{old}}^t\}, \tag{14}$$

$$\mathcal{M}_d = ACC_{\text{new}}^T. \tag{15}$$

However, old classes at different stages are changing and expanding. in the above definitions, $\mathcal{M}_f$ does not truly quantify the forgetting of the initial classes. On the other hand, $\mathcal{M}_d$ only measures the category discovery performance at the last stage, which overlooks measuring the accuracy of new categories throughout the process. As a result, we re-define these two metrics as follows:

$$\mathcal{M}_f = \max_t \{ACC_{\text{init}}^0 - ACC_{\text{init}}^t\}, \tag{16}$$

$$\mathcal{M}_d = \frac{1}{T} \sum_{t=1}^{T} ACC_{\text{new}}^t. \tag{17}$$

In our metrics, $\mathcal{M}_f$ quantify the forgetting of fixed classes set $\mathcal{C}_{\text{init}}^0$ which is more reasonable, and $\mathcal{M}_d$ measures category discovery of each new classes, which could more comprehensively reflect the ability to cluster new classes. In the main manuscript, we use the re-defined $\mathcal{M}_f$ and $\mathcal{M}_d$ to evaluate models in Table 3.

# E    More Experimental Results

## E.1    Confidence Gap with More Confidence Metrics

Here, similar to the preliminary experiments in Figure 2, we train baseline models and provide more metrics, *i.e.*, maximum softmax probability, maximum logit value, margin and negative entropy, of confidence distribution on new and old classes, as illustrated in Figure 7. The results consistently reveal the severe confidence gap between old and new classes, which is the underlying cause of *prediction bias*.

## E.2    Performance of C-CGCD with Longer Stages

In the main paper, we conduct experiments with 5 continual stages by default. To evaluate models in more realistic scenarios with longer continual learning stages, we provide more detailed results of

Table 9: Performance of 10 continual stages on CIFAR100.

| Methods | Stage | 0 | 1 | 2 | 3 | 4 | 5 | 6 | 7 | 8 | 9 | 10 |
|---|---|---|---|---|---|---|---|---|---|---|---|---|
| VanillaGCD | All | 90.82 | 78.42 | 75.68 | 70.35 | 66.64 | 64.29 | 61.05 | 58.33 | 57.14 | 56.23 | 55.15 |
| | Old | - | 82.86 | 80.65 | 73.52 | 69.09 | 68.10 | 63.32 | 60.42 | 59.20 | 57.93 | 56.75 |
| | New | - | 34.00 | 33.00 | 32.40 | 34.80 | 26.00 | 27.00 | 24.80 | 22.20 | 25.60 | 24.80 |
| MetaGCD | All | 90.82 | 81.07 | 76.55 | 74.26 | 67.64 | 64.45 | 61.58 | 59.13 | 60.13 | 56.91 | 56.51 |
| | Old | - | 84.16 | 80.35 | 77.32 | 70.09 | 67.16 | 63.88 | 61.20 | 61.99 | 58.76 | 58.01 |
| | New | - | 50.20 | 34.80 | 37.60 | 35.80 | 26.60 | 27.00 | 26.00 | 28.60 | 23.60 | 28.00 |
| Happy (Ours) | All | 90.36 | **85.62** | **81.88** | **79.82** | **74.01** | **71.81** | **68.46** | **64.05** | **62.14** | **61.38** | **57.81** |
| | Old | - | **85.46** | **81.67** | **79.53** | **76.60** | **73.06** | **71.12** | **66.53** | **63.58** | **61.74** | **59.36** |
| | New | - | **87.20** | **84.20** | **83.20** | **40.40** | **54.40** | **28.60** | **24.40** | **37.80** | **54.80** | **28.40** |

Table 10: Performance of 10 continual stages on TinyImageNet.

| Methods | Stage | 0 | 1 | 2 | 3 | 4 | 5 | 6 | 7 | 8 | 9 | 10 |
|---|---|---|---|---|---|---|---|---|---|---|---|---|
| VanillaGCD | All | 84.20 | 65.15 | 64.63 | 60.94 | 59.46 | 56.52 | 55.47 | 51.65 | 50.66 | 49.83 | 48.56 |
| | Old | - | 68.20 | 67.36 | 63.28 | 61.51 | 58.66 | 57.07 | 53.39 | 52.07 | 51.32 | 49.63 |
| | New | - | 34.60 | 34.60 | 32.80 | 32.80 | 26.60 | 31.60 | 23.80 | 26.60 | 23.00 | 28.20 |
| MetaGCD | All | 84.20 | 68.87 | 65.48 | 62.92 | 60.81 | 58.21 | 56.16 | 54.68 | 52.58 | 50.57 | 48.92 |
| | Old | - | 72.00 | 68.24 | 65.13 | 63.02 | 60.23 | 57.96 | 56.46 | 54.21 | 52.02 | 49.85 |
| | New | - | 37.60 | 35.20 | 36.80 | 32.20 | 30.00 | 29.20 | 26.20 | 24.80 | 24.40 | 31.20 |
| Happy (Ours) | All | 85.86 | **80.75** | **76.92** | **73.34** | **69.77** | **66.33** | **62.75** | **57.56** | **54.73** | **53.02** | **50.69** |
| | Old | - | **84.04** | **79.76** | **75.02** | **72.12** | **67.44** | **64.37** | **59.44** | **55.29** | **53.92** | **50.95** |
| | New | - | **47.80** | **45.60** | **53.20** | **39.20** | **50.80** | **38.40** | **27.60** | **45.20** | **36.80** | **45.80** |

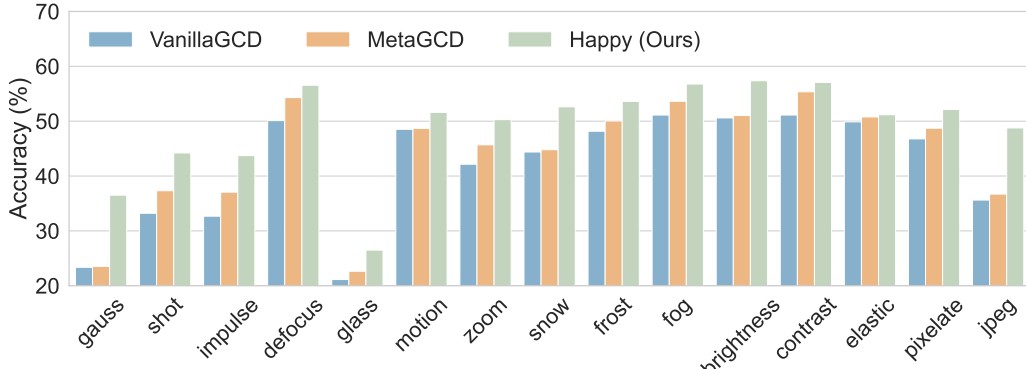

Figure 8: All accuracy on 15 unseen shifted distributions of CIFAR100-C with severity=2.

10-stage C-GCD on CIFAR100 and TinyImageNet, as shown in Table 9 and Table 10. Our method still consistently outperforms others over the whole course of continual stages.

### E.3 Performance under Unseen Distributions

We conduct experiments on the distribution-shift dataset. Specifically, we train models on the original CIFAR100 dataset, and test the model on all 100 classes after 5 stages of training. Models are directly evaluated on the unseen distributions of CIFAR100-C [59], *e.g.*, gaussian_blur, snow and frost, as shown in Figure 8. Our method consistently outperforms others across several unseen distributions, showcasing its strong robustness and generalization ability.

### E.4 Performance under Fine-grained Datasets

Furthermore, we have also conducted experiments on two more fine-grained datasets, *i.e.*, Stanford Cars [60] and FGVC Aircraft [61]. We adopt the default setting of C-GCD described in Section 5.1, *i.e.*, 5 continual stages and 50% of classes serving as $\mathcal{C}_{\text{init}}^0$ initially **labeled** classes. Average accuracies

Table 11: Performance of C-GCD on two more fine-grained datasets.

| Methods | Stanford Cars | | | FGVC Aircraft | | |
|---|---|---|---|---|---|---|
| | All | Old | New | All | Old | New |
| VanillaGCD | 47.00 | 47.73 | 42.61 | 42.95 | 44.35 | 33.38 |
| MetaGCD | 54.67 | 55.28 | 50.95 | 47.16 | 48.61 | 38.23 |
| Happy (Ours) | **62.79** | **63.68** | **57.34** | **53.10** | **53.81** | **48.71** |

over five continual stages are reported in Table 11. Happy also achieves remarkable performance on these fine-grained datasets.

### E.5 Hyper-parameter Sensitivity Analysis

We fix the weights of $\mathcal{L}_{\text{self-train}}$ and $\mathcal{L}_{\text{hap}}$ as 1, considering they are the main objectives for new and old classes. As a result, our method mainly contains three loss weights $\lambda_1, \lambda_2, \lambda_3$ for $\mathcal{L}_{\text{entropy-reg}}$, $\mathcal{L}_{\text{kd}}$ and $\mathcal{L}_{\text{con}}^u$, respectively. Here, we give a sensitivity analysis on CIFAR100 and report average 'All' Acc in Table 12. As shown above, the model is relatively insensitive to $\lambda_3$, whereas $\lambda_1$ and $\lambda_2$ have a more significant impact. Overall, the optimal values for each hyper-parameter are close to 1. In our experiments, we simply set all weights to 1, which shows remarkable results across all datasets. Thus, our method does not require complex tuning of parameters and exhibits strong generalization capabilities and practicability.

Table 12: Sensitivity analysis of hyper-parameters $\lambda_1, \lambda_2$ and $\lambda_3$.

| $\lambda_1$ | 0 | 0.5 | 1.0 | 3.0 | 5.0 |
|---|---|---|---|---|---|
| Acc. | 60.60 | **69.04** | 69.00 | 63.98 | 59.23 |

| $\lambda_2$ | 0 | 0.5 | 1.0 | 3.0 | 5.0 |
|---|---|---|---|---|---|
| Acc. | 65.31 | 66.98 | 69.00 | **69.30** | 68.90 |

| $\lambda_3$ | 0 | 0.5 | 0.7 | 1.0 | 3.0 |
|---|---|---|---|---|---|
| Acc. | 68.74 | 68.94 | **69.16** | 69.00 | 68.92 |

(a) Sensitivity of $\lambda_1$.      (b) Sensitivity of $\lambda_2$.      (c) Sensitivity of $\lambda_3$.

## F  Potential Societal Impacts

This paper focuses on Continual Generalized Category Discovery (C-GCD) and primarily addresses the classification issues. From a more intrinsic perspective, it represents a paradigm of transferring existing knowledge to continuously generalize and learn new information. Therefore, it can be applied to a wide range of tasks and scenarios, such as reasoning abilities in LLMs [62] and multi-modal models [58, 63], continuous pre-training and instruction tuning, and large generative models' generalization abilities to novel concepts. In the fields of biology and health sciences, the principle of C-GCD can assist the discovery of new species and drugs, which will help human beings understand the ecosystem better and facilitate timely diagnosis and treatment of new diseases.

At its core, C-GCD involves leveraging and transferring knowledge learned from old categories to learn new information better, embodying the principle of applying learned concepts to new situations. From this perspective, old knowledge significantly determines the model's ability to discover new knowledge. If biases or unfairness are learned from old knowledge, these issues can also manifest in the newly discovered knowledge. As a result, future works should pay attention to the bias and fairness issues, specifically when learning old classes, and the scrutiny of newly learned knowledge.

