# OpenReview forum: "Happy: A Debiased Learning Framework for Continual Generalized Category Discovery"
_NeurIPS.cc/2024/Conference — NeurIPS 2024 poster_

### Official Review · Reviewer_cVL6 · 2024-06-13

**Soundness:** 3
**Presentation:** 2
**Contribution:** 2
**Rating:** 4
**Confidence:** 3

**Summary:**

The paper presents a novel approach to the task of Continual Generalized Category Discovery (C-GCD). The proposed framework, named Happy, aims to address the challenges of continuously discovering new classes from unlabeled data while preventing the forgetting of previously learned classes. The authors identify two primary issues in C-GCD: prediction bias and hardness bias. To mitigate these, they introduce clustering-guided initialization, soft entropy regularization, and hardness-aware prototype sampling. Experimental results show that the Happy framework significantly improves performance across various datasets, including notable gains on ImageNet-100.

**Strengths:**

Clear Contributions: The paper clearly outlines its contributions, making it easier for readers to understand the novelty and significance of the work.

Experimental Validation: The framework is validated through extensive experiments on multiple datasets, demonstrating significant improvements over state-of-the-art methods. The 7.5% gain on ImageNet-100 is particularly impressive.

Addressing Biases: The identification and mitigation of prediction and hardness biases are well-articulated and experimentally supported. This shows a deep understanding of the underlying issues in continual learning.

**Weaknesses:**

Complexity of the Framework: While the combination of multiple techniques (clustering-guided initialization, soft entropy regularization, and hardness-aware prototype sampling) is innovative, it also adds complexity to the framework. This could make it challenging to implement and tune in practice.

Scalability Concerns: The paper does not extensively discuss the scalability of the proposed framework. As the number of classes and stages increases, the computational and memory requirements could become prohibitive.

Evaluation Metrics: The paper primarily focuses on accuracy improvements but does not delve deeply into other potential evaluation metrics, such as computational efficiency, memory usage, or robustness to different types of data distributions.

**Questions:**

Generalizability to Other Domains: While the framework is validated on multiple vision datasets, it is unclear how well it would generalize to other domains, such as natural language processing or other types of sequential data.

Adapt to the SoTA methods: Is it possible to adopt the method to several significant recent works in the field, such as SPTNet, TIDA, and InfoSieve?

---

> ### Author Rebuttal · Authors · 2024-08-07
>
> Thank you for your insightful advice and valuable questions, we will respond to your concerns point by point.
>
> > W1: Complexity of the Framework.
>
> * **Effect of each component**. Considering that C-GCD is a challenging task, each component is essential and addresses specific issues. (1) Learning new class: Cluster-guided initialization provides a robust initialization for new class heads, while entropy regularization mitigates prediction bias and allocates necessary probabilities for discovering new classes. (2) Preventing forgetting old classes: hardness-aware prototype sampling effectively controls the forgetting of difficult old classes.
> * **Implementation**. (1) Cluster-guided only runs once at each continual stage. (2) Soft entropy regularization, (3) hardness modeling could be implemented on the fly. All of them consume less than 1% of total memory and time, see W3 for more details.
>
> * **Hyper-parameter tuning**.
>   * We fix the loss weights of self-training $L_{self-train}$ and hardness-aware prototype sampling $L_{hap}$ as 1, considering they are the main loss for new and old classes. And our method basically has three weight parameters $\lambda_1,\lambda_2,\lambda_3$ for $L_{entropy-reg}$, $L_{kd}$ and $L_{con}^u$ respectively. Please see **General Response to Common Concern 1** for more details and sensitivity analysis.
>   * Overall, the optimal values for each hyperparameter are close to 1. In our experiments, **we simply set all weights to 1, which shows remarkable results across all datasets.** Thus, our method does not require complex tuning of parameters and exhibits strong generalization capabilities.
>
> > W2: Scalability of the proposed method.
>
> * The time consumption increases linearly with the number of classes and stages, which is the **same for all continual learning methods**.
> * Disk Memory consumption. To mitigate forgetting of old classes, we save one prototype feature per class (768-dim) using about 3KB, whereas previous methods store multiple labeled images, with one image 224× requiring 588KB and ten images requiring over 5000KB. Thus, our method consumes **less than 1/1000** of previous replay-based methods.
>
> > W3: Evaluation Metrics: computational efficiency, memory usage, or robustness to different distributions.
>
> * **Computational efficiency**. The proposed three components are computationally efficient. We run experiments on a single 4090 GPU. For CIFAR100, we train 5 continual stages with 30 epochs for each, it takes $\sim$2.5h in total. (1) Cluster-guided initialization is run only once at the start of each stage, it takes $\sim$15s, only 0.83% of total time. (2) Hardness-aware sampling and (3) entropy regularization occupy less than 1% of the training time. While KD loss takes relatively the most time, around 30% of the time, however, it is necessary and widely used in continual learning.
> * **Disk memory consumption**. As in W2, we only save features for old classes with only <1/1000 memory cost compared with replay-based continual learning. (3KB/5000KB) and our method alleviates privacy issues.
> * **GPU memory consumption**. Training costs around 14192MB GPU memory in total. Within it, hardness-aware sampling and entropy regularization occupy less than 0.1%, and KD loss cost 5672MB, whose main objective is to mitigate forgetting, it consumes even less memory than replay-based methods ($\sim$8000MB) with the same batch size.
> * **Robustness to different distributions**. We have evaluated various methods on several corrupted distributions, see **General Response to Common Concern 2**.
>
> > Q1: Generalizability to Other Domains.
>
> * Good question. This paper primarily explores visual tasks of visual category discovery. The underlying spirit could also be adapted to NLP data, we leave it for future work.
> * For other distributions, we conduct experiments on CIFAR100-C with various corrupted distribution, see **General Response to Common Concern 2**.
>
> > Q2: Adapt to the SoTA methods.
>
> * Great suggestion. We implemented SPTNet [R1] and adapted it to the C-GCD (by incorporating alternate prompt learning), results on CIFAR100 are as follows:
>
>   |   CIFAR100   |    All    |    Old    |    New    |
>   | :----------: | :-------: | :-------: | :-------: |
>   |    SPTNet    |   59.54   |   67.84   |   8.56    |
>   |  SPTNet+LwF  |   63.89   |   70.33   |   24.31   |
>   |    Happy     |   69.00   |   71.82   |   51.36   |
>   | SPTNet+Happy | **70.81** | **73.58** | **54.18** |
>
>   we also adapt InfoSieve [R2] (by incorporating self-supervised code extraction into contrastive learning) as you suggested, results on CUB are shown below:
>
>   |       CUB        |    All    |    Old    |    New    |
>   | :--------------: | :-------: | :-------: | :-------: |
>   |    InfoSieve     |   57.23   |   65.28   |   7.75    |
>   |  InfoSieve+LwF   |   63.18   |   69.30   |   25.60   |
>   |      Happy       |   68.88   |   71.29   |   53.13   |
>   | InfoSieve +Happy | **69.80** | **72.32** | **54.30** |
>
>   Results indicate that adapting SoTA GCD methods to C-GCD with Happy could further enhance the performance.
>
> * We will cite related papers and add them to the revised manuscript.
>
>
>
> **References**:
>
> [R1]. SPTNet: An Efficient Alternative Framework for Generalized Category Discovery with Spatial Prompt Tuning. ICLR 2024.
>
> [R2]. Learn to Categorize or Categorize to Learn? Self-Coding for Generalized Category Discovery. NeurIPS 2023.

---

> > ### Comment · Reviewer_cVL6 · 2024-08-12
> >
> > I read all the comments of the other reviewers. I agree with all weaknesses of the other Reviewers. I hold the same opinions.

---

> > > ### Author Response · Authors · 2024-08-12
> > > **Further Response to Reviewer cVL6**
> > >
> > > Dear Reviewer cVL6,
> > >
> > > Thanks for your feedback. As Reviewer TLvc and jUDg mentioned, our responses have addressed their concerns. Additionally, we have further explained some of your concerns in our previous responses. If you still have some specific questions, please let us know. We are willing to discuss further.
> > >
> > > Thank you for your participation.

---

> ### Author Response · Authors · 2024-08-12
> **Response to Reviewer cVL6**
>
> Dear Reviewer cVL6,
>
> Thanks for your feedback. Considering the "weaknesses of the other Reviewers", we have given responses point by point in the rebuttal. Here, we summarize our responses to the weaknesses pointed out by all reviewers:
>
> * **Hyper-parameters tuning.** Please refer to **General Response to Common Concern 1** and **response to you in W1**. Our method requires three hyper-parameters, and **we only need to set all weights to 1**, which is generalizable and could perform well across all datasets. Therefore, the parameter tuning is simple and easy to implement.
>
> * **Complexity of the method.**
>
>   * Firstly, we have demonstrated the necessity of each module, as shown in Table 5 of the main manuscript.
>   * Secondly, we have provided the computational overhead for our proposed modules (see **responses to you in W2 and W3**). The cluster-guided initialization, entropy regularization, and hardness modeling that we propose consume less than 1% of time and memory resources, which means that they are very efficient and effective.
>
> * **Scalability of the method.** All continual learning methods experience a linear increase in computational demand with the number of classes and stages. Compared to previous replay-based methods, our approach saves disk storage by a factor of 1/1000, as detailed in the **response to you in W2**.
>
> * **Generalization abilities of the method.**
>
>   * We have conducted evaluations on 15 corrupted and shifted distributions of CIFAR100-C, as in the **General Response to Common Concern 1** and **Rebuttal PDF**.
>
>   * Furthermore, previously, we have also conducted experiments on two more fine-grained datasets: **Stanford Cars** and **FGVC Aircraft** datasets, with the results as follows:
>
>     | Standford Cars |    All    |    Old    |    New    |
>     | :------------: | :-------: | :-------: | :-------: |
>     |      GCD       |   47.00   |   47.73   |   42.61   |
>     |    MetaGCD     |   54.67   |   55.28   |   50.95   |
>     |  Happy (Ours)  | **62.79** | **63.68** | **57.34** |
>
>     | FGVC Aircraft |    All    |    Old    |    New    |
>     | :-----------: | :-------: | :-------: | :-------: |
>     |      GCD      |   42.95   |   44.35   |   33.38   |
>     |    MetaGCD    |   47.16   |   48.61   |   38.23   |
>     | Happy (Ours)  | **53.10** | **53.81** | **48.71** |
>
>   * To conclude, our method achieves remarkable performance compared to the previous sota on 6 datasets and 15 unseen distributions, showcasing the strong generalization abilities of Happy.
>
> * **Adapting to SoTA  methods.** We have adapted our method to two sota methods: SPTNet and InfoSieve. Please refer to **response to you in Q2** for more details. We will add all results of adapting Happy to all three methods you listed in the final version. **We will also add citations of the three methods: SPTNet TIDA, and InfoSieve in our reference list.**
>
>
>
> Thank you again for your feedback. We hope our responses and extensive experiments have addressed your concerns. If you have any further questions, we welcome continued discussion. Thank you once again for your review.

---

> > ### Author Response · Authors · 2024-08-13
> >
> > Dear Reviewer cVL6,
> >
> > Thanks for your feedback. As Reviewer TLvc and jUDg mentioned, our responses have addressed their concerns.
> >
> > We were wondering if our latest responses have addressed your concerns. If you have further specific questions on the weaknesses of the method, feel free to raise them and we are willing to discuss them with you. We look forward to and appreciate your feedback.
> >
> > Thank you once again.

---

> > > ### Comment · Reviewer_cVL6 · 2024-08-14
> > >
> > > Thanks for the authors' response. The lack of a solid theoretical foundation and not quite stable debias mechanism affect model reliability. I'm also not quite sure how much hyper-parameters tuning would further affect its stability. Even though they're all ones, but these may coincidentally be optimal.

---

> > > > ### Author Response · Authors · 2024-08-14
> > > > **Further response to Reviewer cVL6**
> > > >
> > > > Dear Reviewer cVL6,
> > > >
> > > > Thank you for your feedback. We will respond to your concerns point by point.
> > > >
> > > > * Lack of a solid theoretical foundation.
> > > >
> > > >   * Actually **we have included the theoretical foundation of our method**. Please refer to **Further Response to Reviewer TDQJ**, we also include this point as below.
> > > >   * The theoretical foundation of our method is the **InfoMax principle** [R1], namely **maximizing the mutual information (MI) between the inputs and outputs of a system.** Here inputs and outputs denotes the feature $Z$ and model predictions $Y$.
> > > >   * For the self-training problem of learning new classes in C-GCD, we employ the maximizing the mutual information (MI) theory. Specifically, the objective can be expressed as $\max I(Y, Z) = H(Y) - H(Y|Z)$, where $Y$ represents the labels and $Z$ the features, with $I$ and $H$ denoting mutual information and entropy, respectively.
> > > >   * The proposed entropy regularization on model predictions $Y$ is the minus of first term $-H(Y)$, while the self-distillation objective aims to encourage model to predict more confidently, resulting in lower prediction entropy, namely the second term of minimizing $H(Y|Z)$, namely maximizing $-H(Y|Z)$.
> > > >   * Overall, our method follows the Infomax principle and aims to maximize mutual information, to ensure the quality of self-learning.
> > > >
> > > > * Not quite stable debias mechanism affect model reliability.
> > > >   * Instability is an inherent issue in C-GCD; however, our approach has effectively mitigated this instability, exhibiting significantly improved stability compared to previous methods.
> > > >   * **Example 1:** On CIFAR100, our method improves the hardest accuracy from 65.10% to 70.23%, and the variance decreases from 23.04% to 10.33%, which means our method greatly mitigates the forgetting of hard old classes and alleviates the extent of imbalance caused by hardness bias.
> > > >   * **Example 2:** Our method exhibits significantly less variability compared to others,On CIFAR100, the maximum new accuracy difference across stages is $\Delta_{new}$=6.3 (56.1-49.8), which is less than the 17.3 (48.9-31.6) of **previous sota** MetaGCD. Similar patterns can be observed in other datasets as well.
> > > >   * Our method is robust and generalizes well to 15 unseen distributions, please see **Rebuttal PDF**.
> > > >   * Overall, our method is a relatively robust method which is much more stable than previous methods.
> > > >
> > > > * Hyper-parameter tuning.
> > > >   * As we responded in **General Response to Common Concern 1** and **Further Response to Reviewer TLvc**, we did not claim that the hyper-parameters are all optimal at 1, but rather, they are **near-optimal**, which is sufficient to achieve SOTA  results across 6 datasets and 15 unseen distributions, without the need for meticulous tuning, thus saving on computational costs.
> > > >   * For $L_{new}$ and $L_{old}$, our results show that a 1:1 weight ratio is optimal, because both old and new classes matter and we need to maintain a balanced focus on both new and old classes as a whole (other weight ratios of $L_{new}$ and $L_{old}$ will result in a 5% to 10% performance degradation). So keep the weight ratio of $L_{new}$ and $L_{old}$ to 1, and the only parameter for tuning is $\lambda$ for $L_{con}^u$. We found that $\lambda$ equals 0.7$\sim$1 works fine. Therefore, It is simple to choose hyper-parameters for our method, just tune $\lambda$ for $L_{con}^u$, which generalizes well on all datasets.
> > > >   * Overall, just keep the weights of old and new loss to 1, and the weight for $L_{con}^u$ be slightly less than 1 (e.g, 0.7), our methods consistently achieves SOTA on all datasets. As a result, the hyper-parameter tuning is relatively simple.
> > > >
> > > >
> > > >
> > > > **References:**
> > > >
> > > > [R1]. Self-organization in a perceptual network. Ralph Linsker. 1988.
> > > >
> > > >
> > > >
> > > > Thanks for your feedback. If you still have questions, please feel free to continue the discussion.

---

### Official Review · Reviewer_jUDg · 2024-07-01

**Soundness:** 4
**Presentation:** 4
**Contribution:** 3
**Rating:** 6
**Confidence:** 5

**Summary:**

The paper proposes a novel method for the Continual Generalized Category Discovery task, addressing the challenges of discovering new classes and preventing forgetting. The approach introduces Clustering-guided Initialization and Group-wise Soft Entropy Regularization for class discovery, as well as Hardness-aware Prototype Sampling for mitigating forgetting. Rigorous experimentation across several datasets demonstrates a significant improvement in performance.

**Strengths:**

1. The organization, presentation, and writing of the paper are very clear, and the figures are attractive and easy to understand.

2. The paper analyzes and proposes solutions for several important issues in Continual Generalized Category Discovery (C-GCD), demonstrating strong innovation.

3. The experimental analysis is thorough, with excellent results. The sufficient ablation studies confirm the contribution of each proposed innovation.

**Weaknesses:**

1. In the C-GCD setting described in the paper, during the continuously discovering stage, the samples of old classes are available. Why not use these old class data directly instead of the Hardness-aware Prototype Sampling method? Some explanation or experimental validation should be added.

2. How is the margin probability computed in line 173 if all the labels are not available?

3. The proposed method uses DINO pretrained ViT-B/16 as the backbone, which, as I understand, is pretrained on the ImageNet dataset. This results in information leakage for CIFAR-100, ImageNet-100, and TinyImageNet datasets in the continual learning process, as all classes are already known in the pretrained model.

**Questions:**

Please see the Weaknesses

---

> ### Author Rebuttal · Authors · 2024-08-07
>
> Thank you for your insightful advice and valuable questions, we will respond to your concerns point by point.
>
> > W1: Why not use these old class data directly instead of hardness-aware prototype sampling?
>
> * Good question. At each continual stage of C-GCD, all training data are unlabeled, i.e., **unlabeled** old and unlabeled new classes samples are mixed together, and the model does not know which samples belong to old categories, thus they cannot be directly utilized.
> * We employ hardness-aware prototype sampling to prevent forgetting, without the need to store class-wise labeled samples of old classes.  By contrast, previous methods store class-wise labeled data, which could cause privacy issues.
>
> > W2: How to compute margin prob in line 173 without labels?
>
> * For the marginal probability, we compute the average of model **predictive** probabilities across a batch, namely $\overline p\in\mathbb{R}^{K^t}=\frac{1}{|B|}\sum_{i\in B}p_i$, which does not require any labels, and **only uses model predictions**.
>
> > W3: DINO pretrained ViT-B/16 backbone results in information leakage for CIFAR100 and TinyImageNet.
>
> * Very good question. We will address this point in detail and incorporate them into the revision.
> * Firstly, DINO is an **self-supervised (unsupervised)** pre-training scheme without any labels. So there is no label leakage in DINO. The main objective of DINO is to learn good and general initial representations for the visual encoder. DINO does not directly learn classification heads for downstream tasks.
>
> * Second, since using a DINO-pretrained ViT has become a standard practice in the literature of GCD [R1, R2, R3], we just followed this setup to ensure a fair comparison.
>
> * Third, we acknowledge your concerns. To thoroughly eliminate the influence of information leakage, we adopted the settings in [R4, R5] and conducted experiments using a DeiT pretrained on 611 ImageNet categories (**explicitly excluding those from CIFAR and TinyImageNet**). Average accuracy over 5 continual stages are shown below:
>
>   |             |           | CIFAR100  |           |           | TinyImageNet |           |
>   | :---------: | :-------: | :-------: | :-------: | :-------: | :----------: | :-------: |
>   |             |    All    |    Old    |    New    |    All    |     Old      |    New    |
>   | VanillaGCD  |   42.50   |   44.57   |   28.16   |   29.28   |    30.04     |   24.02   |
>   |   MetaGCD   |   46.64   |   48.63   |   32.96   |   35.24   |    36.48     |   26.80   |
>   | Happy(Ours) | **69.34** | **71.37** | **57.44** | **53.50** |  **55.67**   | **39.40** |
>
>   As it shows, in scenarios without any information leakage, our method still consistently outperforms previous methods by a large margin. In contrast, methods like MetaGCD, which rely on non-parametric clustering, suffer significant performance declines due to unstable clustering with less pretrained classes.
>
>
>
>
>
> **References**:
>
> [R1]. Generalized Category Discovery. CVPR 2022.
>
> [R2]. Parametric Classification for Generalized Category Discovery: A Baseline Study. ICCV 2023.
>
> [R3]. MetaGCD: Learning to Continually Learn in Generalized Category Discovery. ICCV 2023.
>
> [R4]. Learnability and algorithm for continual learning. ICML 2023.
>
> [R5]. Class Incremental Learning via Likelihood Ratio Based Task Prediction. ICLR 2024.

---

> > ### Comment · Reviewer_jUDg · 2024-08-11
> >
> > Thank you for your response. Some of my concerns have been addressed. However, regarding W2, the author may have misunderstood my question. What I want to know is how to determine whether a sample belongs to an old class or a new class within a session when all labels are unavailable. The marginal probability for old and new classes can only be computed individually if there is a way to accurately separate old class samples from new class samples in each session in an unsupervised manner.

---

> > > ### Author Response · Authors · 2024-08-11
> > > **Response to Reviewer jUDg**
> > >
> > > Dear Reviewer jUDg,
> > >
> > > Thank you very much for your feedback. Here, we provide a detailed explanation of W2: how to compute the marginal probability:
> > >
> > > * In this paper, marginal probability for old $\overline p_{old}$ and new classes $\overline p_{new}$ refer to the sum of  $\overline{\boldsymbol{p}}\in R^{K}$ (marginal probability of all samples in a batch) at the indices of new and old classes, which is calculated **along the class dimension**, **rather than along the sample dimension**.
> > > * Here both $\overline p_{old}$ and $\overline p_{new}$ are **scalars**, while $\overline{\boldsymbol{p}}\in R^{K}$ is a K-dim **vector**, $K=K_{old}+K_{new}$ denotes the number of total classes of the current stage.
> > > * As a result, **we do not need to separate old class samples from new class samples when we compute $\overline p_{old}$ and $\overline p_{new}$**.
> > > * Specifically, for a batch of samples, let $\boldsymbol p_i$ denotes the prediction of the i-th sample. We first average the model predictions for **all samples** along the batch dimension, **without distinguishing between new and old classes samples**, namely $\overline{\boldsymbol{p}}=\frac{1}{|B|}\sum_{i\in B}\boldsymbol p_i$. Here, both $\overline{\boldsymbol{p}}$ and $\boldsymbol p_i$ are $K$ dim vectors. Then we calculate the marginal probabilities for old and new classes **along the class dimension**, i.e., $\overline p_{old}=\sum_{c=1}^{K_{old}}\overline{\boldsymbol{p}}[c]$ and $\overline p_{new}=\sum_{c=K_{old}+1}^{K}\overline{\boldsymbol{p}}[c]$ (so both $\overline p_{old}$ and $\overline p_{new}$ are scalars and $\overline p_{old}+\overline p_{new}=1$). Note that  $\overline{\boldsymbol{p}}[c]$ represents the c-th index of the vector $\overline{\boldsymbol{p}}$.
> > > * **Here is one example**. On the first stage of CIFAR100, there are 50 old classes and 10 new classes, so the dimension of model predictions is $K=60$. We first compute the average of model predictions for all samples in a batch and obtain the **vector** $\overline{\boldsymbol{p}}$. Then we sum the first 50 dimensions of $\overline{\boldsymbol{p}}$ to obtain the **scalar** $\overline p_{old}$ and sum the last 10 dimensions to obtain the **scalar** $\overline p_{new}$.
> > > * To conclude, $\overline p_{old}$ and $\overline p_{new}$ are marginal probabilities of **all** samples **along the dimension of old/new classes**, rather than the marginal probability of **old/new samples**. So we do not need to determine whether a sample belongs to an old class or a new class.
> > >
> > > We really appreciate your feedback. If you have any additional questions, please let us know. We greatly respect your insights and enjoy the discussions with you. Thanks again!

---

> > > > ### Comment · Reviewer_jUDg · 2024-08-12
> > > >
> > > > Thank you for your response. It has addressed some of my concerns, so I will raise my rating to 6.

---

> > > > > ### Author Response · Authors · 2024-08-12
> > > > > **Thanks for your recognition**
> > > > >
> > > > > Dear Reviewer jUDg,
> > > > >
> > > > > We are pleased that our responses have addressed your concerns. Thank you very much for your insightful suggestions and valuable efforts, which are crucial for enhancing the quality of our paper.
> > > > >
> > > > > Thank you once again.

---

### Official Review · Reviewer_TLvc · 2024-07-10

**Soundness:** 3
**Presentation:** 4
**Contribution:** 2
**Rating:** 5
**Confidence:** 4

**Summary:**

The paper points out two-bias issues in CGCD: prediction bias in probability space and hardness bias in feature space. To tackle those two issues, they propose cluster-guided initialization and soft entropy regularization to mitigate prediction bias, and they propose hardness-aware prototype sampling to mitigate hardness bias and forgetting.

**Strengths:**

* The paper is easy to follow. I really appreciate this writing idea: explain the problem and solve the problem.
* The experiment show great improvement over existing method.

**Weaknesses:**

1. Overclaim. The paper argues that they extend CGCD to realistic scenarios. But from the experiment setup, I am not convinced that the scenarios they consider are more realistic than existing work [16,18]. The stage and new classes are still limited.
2. Too many hyper-parameters and loss terms. There are too many hyper-parameters to adjust different loss weights, hindering its generalization. In addition, there are too many loss terms, hindering the understanding of the method. For example, what is the effect of the proposed Eq 4?
3. In Table 5, why KD improve New class by a large margin? What does the new class mean? The total novel class or the new class in the last task?
4. As the paper points out two issues: prediction bias and hardness bias, and proposes methods to mitigate the two issues, they should provide evidence to show that two issues are mitigated.


Minor:
1. The rightmost one in Figure 2 is (d) instead of (b).
2. $L_{hap}$ in eq 10 and 11 should be consistent.
3. citation 23 is a little strange. It seems that it was published in ICCV2023. The citation should be corrected.

**Questions:**

See weaknesses, especially 2 and 4. I will raise my score if the 2 and 4 are well resolved.

---

> ### Author Rebuttal · Authors · 2024-08-06
>
> Thank you for your insightful advice and valuable questions, we will respond to your concerns point by point.
>
> > W1: Overclaim the CGCD setup.
>
> * **Differences from [16,18]**.  We consider (1) more continual stages (5>3 and 10>3 in table 4) with more novel classes to discover (50%>30% of total classes are new) (2) rehearsal-free, i.e., we do not save class-wise labeled samples for old classes, which alleviate privacy and memory issues.
> * Thanks for pointing out the issue. We acknowledge that the settings of our study still have some gaps compared to real-world scenarios and it is a basic exploration towards real-world applications.
> * We will revise the statement as you suggested.
>
> > W2: Too many hyper-parameters and loss terms.
>
> * **Explain of each loss function**. Overall, our method can be divided into three parts: $L_{Happy}=L_{new}+L_{old}+\lambda_3L_{con}^u$. Each of them is important, see table 5 for ablation studies.
>   * $L_{new}$: discover new classes (cluster). It has two parts (1) self-distillation $L_{self-train}$ with sharpen targets for self-training. (2) soft entropy regularization $L_{entropy-reg}$ to alleviate prediction bias, wherein eq (4) is regularization between old and new classes as a whole.
>   * $L_{old}$: mitigating forgetting of old classes. It also has two parts (1) hardness-aware prototype sampling $L_{hap}$ eq (10) and (2) KD loss $L_{kd}$.
>   * $L^u_{con}$: unsupervised contrastive learning (i.e., SimCLR) for basic representation.
>
> * The effect of eq (4) is to reserve the necessary probability for learning new classes and mitigate prediction bias towards old classes.
> * **About hyper-parameters**.
>   * In our method, we fix the weight of $L_{self-train}$ and $L_{hap}$ as 1, considering they are related to the core part of learning new and old classes. (just like cross-entropy loss in standard deep learning)
>   * We only need to tune the weights $\lambda_1,\lambda_2,\lambda_3$ for $L_{entropy-reg}$, $L_{kd}$ and $L_{con}^u$, respectively. For the detailed sensitivity analysis, see the table in **General Response to Common Concern 1**. The optimal weight for the three loss are all near to 1.
>   * In practice, we set all the loss weights to 1 for all datasets without fussy tuning, which works fine and generalizes well to several datasets. As a result, we do not need to extensive search, **just set all the loss weights to 1 and it works fine**.
>
> > W3: In table 5, what does new class mean? Why KD improve new class by a large margin?
>
> * Good question. In table 5, new class means the average accuracy of new classes over 5 stages, i.e., $C_{new}^1,\cdots, C_{new}^5$. i.e., we report the average of new accuracy across five continual tasks.
> * In C-GCD, the classification performance is determined by both the feature space and the classification head.
>   * Without KD loss, the feature space may shift and become mismatched with the head, causing the model to misclassify some new classes as old, which significantly reduces the accuracy of new classes.
>   * Additionally, since the evaluation of C-GCD relies on Hungarian matching across all classes to achieve the highest overall accuracy (see line 224), there are generally more old classes, which tends to stabilize the old classes while new classes experience greater fluctuation.
>
> > W4: Provide evidence to show that two issues are mitigated.
>
> * Very good suggestion. We will append the following results to the revised manuscript.
>
> * **Prediction bias**. We provide two metrics: (1) $\Delta p=\overline p_{old}-\overline{p}_{new}$: the difference in marginal probabilities between old and new classes. (2) the proportion of new classes' samples misclassified as old classes (new$\to$old). The results are as follows (after stage-1):
>
>   |           (in %)           | $\Delta p \downarrow$ (on C100) | new$\to$old $\downarrow$ (on C100) | $\Delta p \downarrow$ (on CUB) | new$\to$old $\downarrow$ (on CUB) |
>   | :------------------------: | :-----------------------------: | :--------------------------------: | :----------------------------: | :-------------------------------: |
>   | Ours w/o $L_{entropy-reg}$ |              81.50              |               63.25                |             83.20              |               65.80               |
>   | Ours w/ $L_{entropy-reg}$  |            **5.76**             |             **10.20**              |           **10.25**            |             **11.05**             |
>
>   The results from two datasets demonstrate that $L_{entropy-reg}$ effectively reduces prediction bias, with a significantly lower marginal probability gap and fewer new class samples misclassified as the old.
>
> * **Hardness bias**. We also present two metrics: (1) $Var_0$: variance in accuracy of the initial labeled classes $C_{init}^0$. (2) hardest Acc: accuracy of the hardest classes in $C_{init}^0$. Results are as follows (after training on 5 stages):
>
>   |           (in %)           | $Var_0 \downarrow$ (on C100) | hardest Acc $\uparrow$ (on C100) | $Var_0 \downarrow$ (on CUB) | hardest Acc $\uparrow$ (on CUB) |
>   | :------------------------: | :--------------------------: | :------------------------------: | :-------------------------: | :-----------------------------: |
>   | Ours w/o hardness sampling |            23.04             |              65.10               |            21.77            |              62.65              |
>   | Ours w/ hardness sampling  |          **10.33**           |            **70.23**             |          **9.28**           |            **68.40**            |
>
>   Results show that hardness-aware sampling effectively reduces hardness bias, with lower accuracy variance and higher hardest Acc.
>
> > Minor typos.
>
> * Thanks for your detailed review. We will correct all the typos carefully.

---

> > ### Author Response · Authors · 2024-08-12
> >
> > Dear Reviewer TLvc,
> >
> > Thanks very much for your time and valuable comments.
> >
> > In the rebuttal period, we have provided detailed responses to all your comments and questions point-by-point for the unclear presentations.
> >
> > Any comments and discussions are welcome!
> >
> >
> >
> > Thanks for your attention and best regards,
> >
> > Authors of Submission 5010.

---

> > ### Comment · Reviewer_TLvc · 2024-08-12
> >
> > I appreciate author's feedback. My concern are resolved. I keep my original score.

---

> > > ### Author Response · Authors · 2024-08-13
> > > **Further Response to Reviewer TLvc**
> > >
> > > Dear Reviewer TLvc,
> > >
> > > We are glad that our responses have resolved your concerns. Regarding the issues **W2 and W4**, which you are particularly concerned about, we have further organized our responses, summarized as follows:
> > >
> > > * W2: Too many hyper-parameters and loss terms.
> > >
> > >   * **Understanding of the method.** Our method consists of three parts: $L_{Happy}=L_{new}+L_{old}+\lambda L_{con}^u$.
> > >
> > >     1. Cluster new classes: $L_{new}$, with self-training and mitigation of prediction bias for new class.
> > >
> > >     2. Mitigate forgetting old classes: $L_{old}$, with hardness-aware prototype sampling to alleviate hardness bias.
> > >
> > >     3. Contrastive learning: $L_{con}^u$ to generally ensure feature representations.
> > >
> > >   * **Hyper-parameter tuning.** For $L_{new}$ and $L_{old}$, our results show that a 1:1 weight ratio is optimal, because both old and new classes matter and we need to maintain a balanced focus on both new and old classes as a whole (other weight ratios of $L_{new}$ and $L_{old}$ will result in a 5% to 10% performance degradation). So keep the weight ratio of $L_{new}$ and $L_{old}$ to 1, and the only parameter for tuning is $\lambda$ for $L_{con}^u$. We found that $\lambda$ equals 0.7$\sim$1 works fine. Therefore, It is simple to choose hyper-parameters for our method, just tune $\lambda$ for $L_{con}^u$, which generalizes well on all datasets.
> > >
> > >   * **Overall.** The three objectives are complementary and non-conflicting, where each is essential, and they collectively improve classification across all classes.
> > >
> > > * W4: Provide evidence to show that two issues are mitigated.
> > >
> > >   * Here, we provide more intuitive examples.
> > >   * **Prediction bias** refers to the bias that the model tends to predict new class samples to the old ones. **Metric:** We compute the proportion of new classes' samples misclassified as old classes. **Example:** In our method, the ratio of new samples misclassified to old ones decreases from 63.25% to 10.20%, which means that Happy could largely mitigate prediction bias, with better separation between old and new classes.
> > >   * **Hardness bias** refers to the bias that models have weaker classification and more severe forgetting on more hard old classes. **Metric:** We compute the accuracy of the hardest class and the accuracy variance among all old classes. **Example:** On CIFAR100, our method improves the hardest accuracy from 65.10% to 70.23%, and the variance decreases from 23.04% to 10.33%, which means our method greatly mitigates the forgetting of hard old classes and alleviates the extent of imbalance caused by hardness bias.
> > >
> > >
> > >
> > > We hope our further responses regarding **W2 and W4** have clearly resolved your concerns. We were wondering whether our paper could be re-evaluated considering these further explanations on **W2 and W4**.
> > >
> > > If you have any other questions, we would be happy to further discuss with you. We really appreciate your feedback.

---

> > > > ### Author Response · Authors · 2024-08-13
> > > >
> > > > Dear Reviewer TLvc,
> > > >
> > > > We were wondering if our latest responses have addressed your concerns about weakness 2 and 4. If you have any other specific questions, feel free to raise them and we are willing to discuss them with you. We look forward to and appreciate your feedback.
> > > >
> > > > Thank you once again.

---

### Official Review · Reviewer_TDQJ · 2024-07-10

**Soundness:** 3
**Presentation:** 3
**Contribution:** 3
**Rating:** 5
**Confidence:** 5

**Summary:**

The article presents a method for Continual Generalized category discovery. A de-biasing learning framework for the Category Discovery (C-GCD) task is designed to address the challenge of continuously discovering new concepts in an ever-changing environment while maintaining recognition of known categories. Traditional C-GCD studies have some limitations, such as storage and privacy issues caused by storing samples of past classes, and only considering limited incremental stages or assuming a proportion of known samples, which are not in line with practical application scenarios. Therefore, the study focuses on a more realistic C-GCD setup that includes more learning phases, more new categories, and after each phase, data from previous phases is not accessible.

**Strengths:**

Originality: In this paper, the authors propose a novel de-biased-learning framework, "Happy, "specifically for the continuous Generalized Category Discovery (C-GCD) task, which is a relatively underexplored research area. It has designed a unique set of methods to handle the challenge of incrementally discovering new categories in unlabeled data while preserving the ability to recognize old categories, which traditional machine learning and deep learning approaches have overlooked. In particular, the proposed cluster-guided initialization, soft entropy regularization and hardness-aware prototype sampling strategies are innovative solutions to the unique problems of C-GCD.
Quality: The quality of the paper is reflected in its detailed theoretical analysis and experimental verification. The authors not only elaborate the design principle and motivation behind the method, but also conduct extensive experiments on multiple datasets to prove the effectiveness of the "Happy" framework achieving significant performance improvement, showing its strong generalization ability and practicality. In addition, the paper also reflects on the limitations and assumptions of the method, indicating that the author has deeply considered the comprehensiveness and rigor of the study.
Clarity: The structure of the paper is clear and the logic is coherent. From the introduction, which clearly explains the research background and motivation, to the method, which analyzes the components of the framework and their working principles in detail, to the presentation and analysis of the experimental results, every step is well organized. In addition, the authors also provide detailed implementation details and algorithm flow in the appendix, which increases the readability and reproducibility of the paper.

**Weaknesses:**

1. Although the authors acknowledge and briefly discuss the social impact of technology, the paper does not detail specific negative impacts that a "Happy" framework could bring, such as potential bias transmission, fairness issues, privacy violations, or security risks. For an algorithm intended for application in the open world, the lack of a comprehensive social impact analysis may limit its acceptance and ethical application in practice.
2. It is clearly stated that this study does not include theoretical results, which means that a complete hypothesis set and theoretical proof are not provided to support the validity of the proposed method. While empirical research has shown the effectiveness of "Happy, "the lack of a solid theoretical foundation may diminish its persuasive power in academia.
3. The authors acknowledge that there are some assumptions and limitations to the study, such as overfitting the model, performance in noisy environments, and testing on specific datasets. These factors may limit the method's general applicability and ability to generalize, especially in different data distributions or more complex real-world environments.
4. It is pointed out that even after the introduction of the debias mechanism, the recognition accuracy of the new class still fluctuates at different incremental stages, which indicates that the model may have stability problems when processing data of different class difficulty. This volatility can affect model reliability and user trust in real-world deployments.

**Questions:**

1. In the experimental part, how do you make sure that in the unsupervised increment phase, the model not only finds new categories, but also accurately distinguishes them without confusing them with the old ones? Are there specific metrics or experimental Settings to measure this ability to differentiate?
2. How does the soft entropy regularization realize the reasonable distribution of the probability of the new class? Can you explain this process in detail and how it is combined with cluster-guided initialization to improve the clustering performance of new categories?
3. In the hardness-aware prototype sampling strategy, how do you define and quantify the "hardness" of a class, and how do you sample effectively against this hardness to mitigate forgetting? Are there concrete examples of how this strategy helps the model remember old categories that are difficult to classify?
4. The paper mentions that in preliminary experiments it was found that the model tends to misclassify new categories into old ones, and that the features of the old categories are disturbed when learning the new ones. In addition to the proposed solutions, have other technologies, such as meta-learning or memory enhancement networks, been considered to further improve these problems?

**Limitations:**

The authors discussed the limitations of the work in the paper, including the confidence calibration problem and the scope of the study mainly focused on the classification task.
Confidence Calibration problem: In the Continuous Generalized Category Discovery (C-GCD) task, the confidence of the model is not calibrated due to an imbalance in the labeling conditions between the initial phase and the sustained phase. This leads to a clear gap between the old category and the new category, and even degrades performance when incorporating prior information.
The scope of application is limited to classification tasks: Although this paper mainly discusses the C-GCD learning paradigm under classification tasks, the application of this method has not been extended to other fields, such as object detection and image segmentation.
The authors note that future work should consider how confidence calibration techniques can be incorporated into C-GCD to further reduce potential bias. At the same time, they encourage the extension of the C-GCD learning framework to more types of visual tasks in order to expand its applicability and impact. These discussions reflect a clear understanding of the limitations of the current findings and suggest possible directions for future research.

---

> ### Author Rebuttal · Authors · 2024-08-06
>
> Thank you for your insightful advice and valuable questions, we will respond to your concerns point by point.
>
> > W1:  Lack of a comprehensive social impact analysis.
>
> * We summarize the negative impacts: (1) **Bias/error accumulation**. Given the stringent unlabeled conditions of C-GCD, the initial learning bias may continuously accumulate, leading to a greater bias in later stages. (2) **Spurious correlations**. Category discovery relies on knowledge learned from labeled classes. If incorrect correlations are learned, e.g., over-reliance on the background, these may interfere with the learning of new ones.
>
> * We will discuss them in detail in the final version.
>
> >  W2: Lack of a theoretical foundation.
>
> * Happy is essentially self-learning on unlabeled data, and the theoretical foundation is conditional entropy maximization.
> * Specifically, the objective can be expressed as $\max I(Y, Z) = H(Y) - H(Y|Z)$, where $Y$ represents the labels and $Z$ the features, with $I$ and $H$ denoting mutual information and entropy, respectively. The first term is equivalent to the proposed marginal entropy regularization $L_{entropy-reg}$, while the second one $H(Y∣Z)$, can be computed by the posterior probability $p(y|z)$, which we aim to minimize through self-distillation $L_{self-train}$.
>
> * We will include this theoretical analysis.
>
> > W3: There are assumptions that may limit the general applicability especially in different distributions.
>
> * Our method achieves sota not only on four datasets (table 2) but also on the distribution-shift CIFAR100-C. (**See General Common Concern 2 for more details**). The results demonstrate that our method has a stronger generalization ability.
>
> > W4: The accuracy of the new class still fluctuates at different incremental stages with the debias methods.
>
> * The fluctuation in new categories is an inherent issue in C-GCD. However, our method exhibits significantly less variability compared to others, e.g., on CIFAR100, the maximum new accuracy difference across stages is $\Delta_{new}=6.3$ (56.1-49.8), which is less than the 17.3 (48.9-31.6) of MetaGCD. Similar patterns can be observed in other datasets as well.
>
> > Q1: How to make sure the model not only finds new categories but also avoids confusing them with the old ones? Are there specific metrics?
>
> * We follow SimGCD and employ a unified classification head for both old and new classes in a joint prediction space. So our method could classify both new and old categories together, which not only implicitly differentiates between new and old classes but also discovers new ones.
>
> * **Specific metrics**. We calculate the proportion of new class samples misclassified as old classes (new$\to$old) and old classes misclassified as new (old$\to$new) after stage-1 training. Results are shown below:
>
>   | misclassify ratio | new$\to$old (C100) | old$\to$new (C100) | new$\to$old (CUB) | old$\to$new (CUB) |
>   | :---------------: | :----------------: | :----------------: | :---------------: | :---------------: |
>   |      SimGCD       |       85.75        |        6.78        |       84.18       |       6.65        |
>   |      MetaGCD      |       30.65        |        6.64        |       32.36       |       6.50        |
>   |       Happy       |     **10.20**      |      **4.35**      |     **11.05**     |     **4.95**      |
>
>   Results show Happy can more effectively distinguish between new and old classes with a lower misclassify ratio, and achieves a higher accuracy for new classes (e.g., 51.3>31.6 at stage-5 in table 2).
>
> > Q2: Explain soft entropy regularization and cluster-guided initialization.
>
> * Soft entropy regularization. See Sec 3.2 line 128, as the model tends to bias the predictions to old classes, so we implement an explicit constraint to ensure necessary predictive probabilities for new classes to facilitate learning. Considering that at each stage the prior ratio of old and new classes is inaccessible, we employ a soft regularization, assuming a general balance between the probabilities for new and old classes ($p_{old} = p_{new} = 0.5$), which is achieved by the marginal entropy maximization.
>
> * Cluster-guided initialization ensures a good **initialization** for new class heads, while entropy regularization improves **training** of new classes with less prediction bias.
>
> > Q3: How to define "hardness" and how to sample against this hardness? Concrete examples of how it helps remember difficult old classes?
>
> * As in Sec 3.4 Eq (9), we define class-wise **hardness** $h_c$ as the average cosine similarity between the class center $\mu_c$ and other class center $\mu_j$, where $\mu_j$ is computed in Eq (8). Intuitively, the greater the similarity to other classes, the more likely it is that its feature space will be confused with others, thereby the class is more difficult.
>
> * To sample against hardness, considering more difficult categories tend to be forgotten more readily, we sample harder classes more frequently. So we treat $h_c$ as logits and model hardness distribution with softmax function in Eq (9).
>
> * **Concrete samples**. After learning 5 stages, we evaluate the model on initial labeled classes $C_{init}^0$ and the accuracy of the hardest class (lowest acc) is reported:
>
>   | hardest Acc |   C100    |    CUB    |
>   | :---------: | :-------: | :-------: |
>   |   SimGCD    |   62.85   |   59.68   |
>   |   MetaGCD   |   65.31   |   61.26   |
>   |    Happy    | **70.23** | **68.40** |
>
>   Results show that Happy could remember difficult classes better with higher hard class accuracy.
>
> > Q4: Have other technologies been considered to further improve the task?
>
> * MetaGCD actually utilizes meta-learning, but the results are weaker than ours (see table 2), because MetaGCD only simulates the C-GCD process without any debiasing mechanism.
> * In essence, meta-learning and memory enhancement can be integrated with our debiasing mechanism to further enhance the results. We leave this aspect to future work.

---

> > ### Author Response · Authors · 2024-08-12
> >
> > Dear Reviewer TDQJ,
> >
> > Thanks very much for your time and valuable comments.
> >
> > In the rebuttal period, we have provided detailed responses to all your comments and questions point-by-point for the unclear presentations.
> >
> > Any comments and discussions are welcome!
> >
> >
> >
> > Thanks for your attention and best regards,
> >
> > Authors of Submission 5010.

---

> > ### Comment · Reviewer_TDQJ · 2024-08-12
> >
> > Thanks to the author for answering my question, but it failed to solve my doubts about the weakness of the method

---

> > > ### Author Response · Authors · 2024-08-13
> > > **Further Response to Reviewer TDQJ**
> > >
> > > Dear Reviewer TDQJ,
> > >
> > > Thank you very much for your feedback. Regarding the weaknesses you've highlighted, we have further organized our responses and summarized them as follows:
> > >
> > > * W1: Lack of a comprehensive social impact analysis.
> > >   * Here, We have included a more detailed discussion on societal impacts, supplemented with specific examples.
> > >   * **Medical risks:** Our method could be applied in the medical field, where biases in learned categories might affect the discovery or diagnosis of new diseases. Over time, these errors could increase, potentially delaying critical medical interventions.
> > >   * **Fairness issues:** Regarding different genders and populations, our model may learn unfair biases from the labeled data, which could then be transferred to new domains. This could perpetuate inequalities within the model's predictions.
> > >
> > > * W2: Lack of a theoretical foundation.
> > >   * The theoretical foundation of our method is the **InfoMax principle** [R1], namely **maximizing the mutual information (MI) between the inputs and outputs of a system.** Here inputs and outputs denotes the feature $Z$ and model predictions $Y$.
> > >   * For the self-training problem of learning new classes in C-GCD, we employ the maximizing the mutual information (MI) theory. Specifically, the objective can be expressed as $\max I(Y, Z) = H(Y) - H(Y|Z)$, where $Y$ represents the labels and $Z$ the features, with $I$ and $H$ denoting mutual information and entropy, respectively.
> > >   * The proposed entropy regularization on model predictions $Y$ is the minus of first term $-H(Y)$, while the self-distillation objective aims to encourage the model to predict more confidently, resulting in lower prediction entropy, namely the second term of minimizing $H(Y|Z)$, namely maximizing $-H(Y|Z)$.
> > >   * Overall, our method follows the Infomax principle and aims to maximize mutual information, to ensure the quality of self-learning.
> > > * W3: There are assumptions that may limit the general applicability, especially in different distributions.
> > >   * We need to clarify here that our method does not rely on explicit assumptions or predefined ratios of new to old categories in previous methods, thereby enhancing its generalizability. This has been demonstrated in our experiments, including the distribution shift experiments detailed in the Rebuttal PDF.
> > > * W4: The accuracy of the new class still fluctuates, which can affect model reliability and user trust in real-world deployments.
> > >   * Our method significantly mitigates fluctuations. For example, in CIFAR 100, the gap in accuracy among classes has been reduced from 17.3% to 6.3%. This ensures the reliability and stability of our approach in real-world deployments.
> > >
> > >
> > >
> > > **References:**
> > >
> > > [R1]. Self-organization in a perceptual network. Ralph Linsker. 1988.
> > >
> > >
> > >
> > > Hope these further responses address your concerns about the weaknesses part. If you have further questions, please let us know. We are willing to have more discussions with you. Thanks for your participation.

---

> > > > ### Author Response · Authors · 2024-08-13
> > > >
> > > > Dear Reviewer TDQJ,
> > > >
> > > > We were wondering if our latest responses have addressed your concerns. If you have any other specific questions, feel free to raise them and we are willing to discuss them with you. We look forward to and appreciate your feedback.
> > > >
> > > > Thank you once again.

---

### Author Rebuttal · Authors · 2024-08-06

We thank all reviewers for their dedication and insightful comments, and we believe these comments are significant for improving the overall quality of this paper.

We are pleased that the reviewers appreciate our paper from various aspects, including the novelty of the method [TDQJ,jUDg,cVL6], clear writing [TLcv, jUDg, cVL6], and remarkable performance [TDQJ, TLvc, jUDg, cVL6].

In this paper, we explore the task of Continual Generalized Category Discovery (C-GCD). At each continual stage, data from new and old classes are mixed together without labels. Compared to previous settings, our setup is more realistic, (1) encompassing more stages and new classes to discover, (2) does not preserve class-wise labeled samples of old classes to replay.

Here, we respond to some common concerns:

> Common Concern 1: The method is complex with many hyper-parameters.

* Our approach can be divided into three parts: (1) $L_{new}$: employs self-distillation $L_{self-train}$ and $L_{entropy-reg}$ to reduce prediction bias to learn new classes. (2) $L_{old}$: hardness-aware prototype sampling $L_{hap}$ and $L_{kd}$ to further mitigate hardness bias and forgetting of old classes. (3) $L_{con}^u$: contrastive learning to ensure basic representations. Each of them is important.
* **Hyper-parameters**. We fix the weight of $L_{self-train}$ and $L_{hap}$ as 1, considering they are the main objectives for new and old classes. As a result, our method mainly contains three loss weights $\lambda_1,\lambda_2,\lambda_3$ for $L_{entropy-reg}$, $L_{kd}$ and $L_{con}^u$ respectively. Here, we give sensitivity analysis on CIFAR100 as follows: (average All Acc over 5 stages)

  | $\lambda_1$ ($L_{entropy-reg}$) |   0   |    0.5    |  1.0  |  3.0  |  5.0  |
  | :---------------------------------------: | :---: | :-------: | :---: | :---: | :---: |
  |                  All Acc                  | 60.60 | **69.04** | 69.00 | 63.98 | 59.23 |

  | $\lambda_2$ ($L_{kd}$) |   0   |  0.5  |  1.0  |    3.0    |  5.0  |
  | :------------------------------: | :---: | :---: | :---: | :-------: | :---: |
  |             All Acc              | 65.31 | 66.98 | 69.00 | **69.30** | 68.90 |

  | $\lambda_3$($L_{con}^u$) |   0   |  0.5  |    0.7    |  1.0  |  3.0  |
  | :--------------------------------: | :---: | :---: | :-------: | :---: | :---: |
  |              All Acc               | 68.74 | 68.94 | **69.16** | 69.00 | 68.92 |

  As shown above, the model is relatively insensitive to $\lambda_3$, whereas $\lambda_1$ and $\lambda_2$ have a more significant impact. Overall, the optimal values for each hyperparameter are close to 1.

  In our experiments, **we simply set all weights to 1 which shows remarkable results across all datasets.** Thus, our method does not require complex tuning of parameters and exhibits strong generalization capabilities.

* **Computational costs**. The proposed three components: cluster-guided initialization, hardness-aware sampling and entropy regularization bring very small computation overhead, and the consumption (both time and GPU memory) of each part is **less than 1%**. More details are in **Response to Reviewer [cVL6]**.

> Common Concern 2: Generalization to different distributions.

* We conduct experiments on the distribution-shift dataset CIFAR100-C [R1] with severity=2, and test the model on all 100 classes after 5 stages of training. The results are shown below:

  | Distribution | Original  | +gauss noise |   +snow   |   +fog    | +motion blur | +pixelate |
  | :----------: | :-------: | :----------: | :-------: | :-------: | :----------: | :-------: |
  |  VanillaGCD  |   51.36   |    23.36     |   44.38   |   51.12   |    48.52     |   46.77   |
  |   MetaGCD    |   55.78   |    23.53     |   44.80   |   53.63   |    48.69     |   48.72   |
  | Happy (Ours) | **59.99** |  **36.51**   | **52.61** | **56.77** |  **51.60**   | **52.13** |

  Our method consistently outperforms others across several unseen distributions, showcasing its strong robustness and generalization ability.

* **See the Rebuttal PDF for all 15 distributions of CIFAR100-C**.

> Common Concern 3: Evidence to show prediction bias and hardness bias are mitigated.

* **Prediction bias**. We present two metrics (after stage-1): (1) $\Delta p=\overline p_{old}-\overline{p}_{new}$: the gap of marginal predictive probability on old and new classes, (2) the proportion of new classes samples misclassified as old ones. Both metrics are low in our method, which shows that the bias is mitigated. **See response to Reviewer [TLvc] for more details**.
* **Hardness bias**. We also show two metrics (after stage-5): (1) $Var_0$: the variance of accuracy among the initially labeled classes $C_{init}^0$, (2) $acc_{hard}$: accuracy of the hardest class. The former decreases and the latter increases with our method, which means that the bias is mitigated. **See response to Reviewer [TDQJ, TLvc] for more details**.

For other specific questions, we respond to each reviewer point by point as below. We will carefully revise all comments from the four reviewers and incorporate them into the revised paper. Thanks again to all reviewers for their valuable suggestions!


**References**:
[R1]. Benchmarking Neural Network Robustness to Common Corruptions and Perturbations. ICLR 2019.

---

### Decision · Program_Chairs · 2024-09-25

**Decision:**

Accept (poster)

**Comment:**

This paper studies a new task called Continual Generalized Category Discovery (C-GCD), which aims to incrementally discover more new classes from unlabeled data over a long period while maintaining the ability to recognize previously learned classes without storing samples of past classes. There are two challenges in C-GCD, including the prediction bias in probability space and the hardness bias in feature space. To address these challenges, this paper proposes cluster-guided initialization and soft entropy regularization to mitigate prediction bias, and proposes hardness-aware prototype sampling to mitigate hardness bias and forgetting. Experimental results demonstrate the effectiveness of the proposed method.

This paper finally received 1x Borderline reject, 2x Borderline accept, 1x Weak accept. Reviewer TDQJ has raised concerns about the negative impacts of the proposed method, lack of theoretical results, the limited applicability, and the stability of the method. After the authors' rebuttal, Reviewer TDQJ indicated that some concerns were addressed. Reviewer TLvc thinks that the setting is over-claimed, there are too many hyper-parameters and loss terms, more evidence should be provided to show that the two issues are mitigated. The authors have provided corresponding feedback, and Reviewer TLvc confirmed that these concerns were resolved and would keep the original score (Borderline accept). Reviewer jUDg mainly has concerns about the C-GCD setting and the potential information leakage issue. After discussing with the authors, Reviewer jUDg acknowledged that these concerns were addressed and increased the score from 6 (Weak accept). Reviewer cVL6 (Borderline reject) has concerns about the complexity of the framework, scalability concerns, evaluation metrics, generalizability to other domains, and adaptation to state-of-the-art methods. Reviewer cVL6 has also been involved in the discussion with the authors, but indicated that most of the concerns were not addressed. Reviewer cVL6 still has concerns about the hyperparameters and theoretical contributions. I have checked all the comments from Reviewer cVL6 and the authors, I would agree with Reviewer cVL6 regarding the two concerns, but I feel that the hyperparameter issue is not a fatal problem. Besides, the lack of theoretical contributions also may not be a very strong reason to reject this paper, because this paper studies a new practical task with an innovative method achieving satisfactory performance. I personally think that whether a paper should be accepted should not simply depend on whether this paper has some minor problems. We should focus on the key merits of the paper. As most reviewers acknowledged, this paper indeed has merits in a new task, an innovative method, and good performance.

The current average score of this paper is 5.0, which means that the decision of this paper is clearly on the borderline. I have carefully checked the paper, all the comments, and all the discussions. I noticed that the relatively low score is actually caused by some minor problems. After much consideration, I feel that this paper could be accepted.